# Thickness-aware E(3)-Equivariant 3D Mesh Neural Networks

**Sungwon Kim** [1]  **Namkyeong Lee** [2]  **Yunyoung Doh** [3]  **Seungmin Shin** [3]  **Guimok Cho** [3]
**Seung-Won Jeon** [3]  **Sangkook Kim** [3]  **Chanyoung Park** [1][2]

## Abstract

Mesh-based 3D static analysis methods have recently emerged as efficient alternatives to traditional computational numerical solvers, significantly reducing computational costs and runtime for various physics-based analyses. However, these methods primarily focus on surface topology and geometry, often overlooking the inherent thickness of real-world 3D objects, which exhibits high correlations and similar behavior between opposing surfaces. This limitation arises from the disconnected nature of these surfaces and the absence of internal edge connections within the mesh. In this work, we propose a novel framework, the Thickness-aware E(3)-Equivariant 3D Mesh Neural Network (T-EMNN), that effectively integrates the thickness of 3D objects while maintaining the computational efficiency of surface meshes. Additionally, we introduce data-driven coordinates that encode spatial information while preserving E(3)-equivariance or invariance properties, ensuring consistent and robust analysis. Evaluations on a real-world industrial dataset demonstrate the superior performance of T-EMNN in accurately predicting node-level 3D deformations, effectively capturing thickness effects while maintaining computational efficiency.

## 1. Introduction

Advances in static analysis have become essential across various fields, including structural engineering (Whalen et al., 2021), materials science (Panthi et al., 2007; Wei et al., 2019), and geophysics (Ren & Tang, 2010; Schwarzbach et al., 2011). These analyses enable detailed physics-based predictions, such as deformation, stress distribution, and load testing, which are critical for designing and optimizing

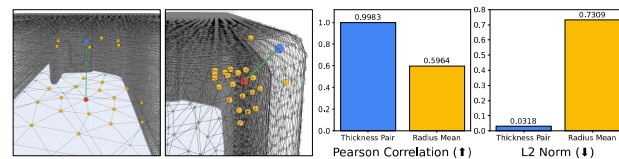

*Figure 1.* The left figures show a mesh, with two different target nodes (●), their thickness paired nodes (●), thickness distance (—), and nearby nodes within a radius (●). The right figures compare Pearson correlation and L2 Norm between the target node's deformation and its thickness paired / nearby nodes within a radius.

complex systems. Traditionally, computational numerical solvers like finite element methods (FEM) (Klocke et al., 2002; Felippa, 2004) have been the primary tools for such tasks. While accurate, these solvers often involve high computational costs and extended runtimes, limiting their scalability for real-time or large-scale applications.

In recent years, mesh-based 3D analysis methods (Pfaff et al., 2020; Suk et al., 2021; Trang et al., 2024) have emerged as a promising alternative to traditional numerical solvers. By employing a graph-based representation of 3D object surfaces, these methods facilitate enhanced computational efficiency while accurately modeling and preserving the intricate surface topology and geometric properties of the objects. However, existing mesh-based methods focus solely on modeling the surfaces of 3D objects, overlooking their *thickness*. Real-world objects such as plates, baskets, and layered materials inherently possess thickness, where interactions between opposing surfaces significantly influence one another. This is because many surface-related physical behaviors (e.g., deformation and bending) are directly influenced by bending, shear, and buckling stiffness, all of which depend on thickness (Okafor & Oguaghamba, 2009; Wang, 2010; Renton, 1991; Gürdal et al., 2008). Without internal interactions to model the empty space enclosed by these surfaces, these methods fail to capture the coupled dynamics and correlations between opposing surfaces, leading to inaccuracies when applied to objects with thickness.

To quantitatively illustrate the significance of these interactions, we present an analysis in Fig. 1, highlighting the strong relationship between opposing surfaces that collectively define the thickness of an object. On these surfaces, we identify two nodes that are positionally aligned on each opposing surface (i.e., with coinciding normal vectors) as a *thickness node pair*. To verify the highly correlated and simi-

[1]Graduate School of Data Science, KAIST, Daejeon, Republic of Korea [2]Industrial & Systems Engineering, KAIST, Daejeon, Republic of Korea [3]LG Electronics, Pyeong-taek, Republic of Korea. Correspondence to: Chanyoung Park <cy.park@kaist.ac.kr>.

*Proceedings of the 42$^{nd}$ International Conference on Machine Learning*, Vancouver, Canada. PMLR 267, 2025. Copyright 2025 by the author(s).

lar behavior of thickness node pair, we compare the Pearson correlation and L2 Norm between the deformation of the target node (● in Fig. 1) and its thickness paired node (●). We also include comparisons in terms of the mean deformation of nodes within a defined radius[1] (●), as existing mesh-based methods rely on radius-based aggregation (Pfaff et al., 2020; Anandkumar et al., 2020). As shown in Fig. 1, thickness node pairs exhibit significantly higher correlation and similarity compared to the mean of nearby nodes within a radius, demonstrating that the modeling relationship among thickness node pairs would be beneficial in accurately modeling the behavior of 3D objects with thickness. However, mesh-based objects, which represent the geometry and topology of surfaces, face challenges in accurately modeling these interactions due to the lack of connections between opposing surfaces within the mesh.

Motivated by these findings, this work presents a novel framework that effectively incorporates the inherent thickness of 3D objects, enhancing mesh-based methods by enabling precise interaction modeling between opposing surfaces while maintaining computational efficiency.

In addition, while considering thickness helps capture geometric properties that influence structural behavior, learning spatial information (i.e., coordinates) is also important, as it enables the accurate representation of the spatial continuity inherent in field variables such as stress distribution and deformation. Therefore, incorporating spatial information in learning-based surrogate models would further enhance their ability to generalize physical behavior patterns by capturing spatial relationships. However, when utilizing spatial information, it is crucial to consider that preserving E(3)-equivariance is essential in static analysis, as transformations such as rotation, translation, and reflection (i.e., the E(3)-group) do not alter material properties.

To ensure the E(3)-equivariance of spatial information in a mesh neural network, we consider leveraging E(3)-equivariant graph neural network approaches. These methods often transform the original coordinate system into higher-dimensional representations or utilize specialized equivariant functions, such as spherical harmonics (Worrall & Brostow, 2018; Gasteiger et al., 2020; Brandstetter et al., 2021; Batatia et al., 2023; Batzner et al., 2022). Although effective, these techniques typically introduce substantial computational overhead, making them less suitable for real-world industrial applications, particularly for large-scale meshes with a high number of nodes and edges. In contrast, methods that avoid such computationally intensive techniques often adopt constrained representations, relying solely on local geometric properties (e.g., relative displacements, cross products) (Satorras et al., 2021; Trang et al., 2024; Du et al., 2022) or excluding directional and

coordinate-based information altogether. However, these simplifications limit their ability to capture global spatial contexts and interactions, which are critical for accurate and comprehensive analysis.

To address these challenges, we employ *data-driven coordinates*, allowing the model to directly use 3D coordinate features as neural network inputs. This approach allows for robust and expressive representations of object geometry while effectively capturing global spatial contexts across the entire shape. Specifically, our proposed data-driven coordinates ensure invariance to E(3)-transformations. This approach guarantees consistent spatial information, achieves computational efficiency, and preserves the richness of spatial representation.

The key contributions of this study are as follows:

- **Thickness-Aware Framework**: We propose a Thickness-aware E(3)-Equivariant 3D Mesh Neural Networks (T-EMNN) that accurately models interactions between opposing surfaces while retaining computational efficiency.

- **Data-Driven Coordinates**: Our framework incorporates data-driven coordinates to ensure consistent and robust spatial representation of 3D objects.

- **E(3)-Equivariance or Invariance**: The proposed model preserves E(3)-equivariance or invariance, ensuring robustness to transformations such as translations, rotations, and reflections.

- **Validation on Real-World 3D Objects**: We validate our approach on real-world 3D objects in industry, demonstrating accurate node-level 3D deformation predictions in practical scenarios.

## 2. Related Work

### 2.1. Mesh-Based 3D Representation

Meshes (Kato et al., 2018; Pfaff et al., 2020; Feng et al., 2019; Trang et al., 2024; Rubanova et al., 2021; Li et al., 2018b) encode both surface and volumetric characteristics through vertices, edges, and faces, facilitating geometric and topological analysis. Recent neural architectures (Smirnov & Solomon, 2021; Li et al., 2023; Singh et al., 2021; Lahav & Tal, 2020) leverage mesh structures for enhanced 3D analysis by incorporating geometric and topological features. For instance, MGN (Pfaff et al., 2020) models physical interactions via graph-based message passing with distances, displacements, and world distances between connected nodes. MeshNet (Feng et al., 2019) captures structural information from mesh faces through kernel-based operations, while Milano et al. (Milano et al., 2020) incorporate both edge and face geometries using attention mechanisms.

Beyond learning mesh geometry and topology, several methods address the limitations of message-passing, where local communication is constrained by mesh density (Gao

---

[1]The radius was set to the average thickness of the shape for consistency.

& Ji, 2019; Han et al., 2022; Fortunato et al., 2022). To enhance global information exchange, hierarchical pooling techniques have been introduced to extend message propagation beyond immediate neighbors (Cao et al., 2022; Han et al., 2022; Janny et al., 2023; Yu et al., 2023).

Furthermore, the geometric and topological richness of mesh graphs has driven research into equivariant representations. Many approaches (Shakibajahromi et al., 2024; Trang et al., 2024) build on E(n)-equivariant graph neural networks (EGNN) (Satorras et al., 2021), leveraging their efficiency for large-scale mesh data. EGNN ensures E(3)-equivariance via message passing while avoiding direct nonlinear spatial encoding. EMNN (Trang et al., 2024) extends EGNN by generating E(3)-invariant messages that incorporate geometric features like face areas and normal magnitudes, enhancing equivariant modeling while preserving efficiency. However, both approaches struggle to capture global spatial relationships, as they rely on relative positions and directional updates rather than fully embedding spatial features.

## 3. Preliminaries

### 3.1. Notations

A shape is represented as a mesh $M = (V, E)$, where the nodes $V$ correspond to unique coordinates in 3D space, and the edges $E$ define the connectivity between nodes. Let the coordinate of a node $v_i \in V$ be denoted as $\mathbf{x}_i \in \mathbb{R}^3$, representing its position in 3D space. In this work, we denote the mesh as the surface mesh, where its nodes and edges are only located on the surface of the shape, ensuring that the mesh is water-tight. Each shape is associated with experimental conditions $C$, which influence the results of simulation. The goal of this study is to predict the deformation of each node along the $x$, $y$, and $z$ axes, given the shape and the experimental condition $\mathbf{c} \in C$ as inputs. Formally, the deformation at a node $v_i$ is represented as $\Delta \mathbf{x}_i = [\Delta x_i, \Delta y_i, \Delta z_i]$, which is the output of the model.

Each face in the mesh is defined as a triangle consisting of three connected nodes, and the outward-facing normal vector of a face $f_k$ is denoted as $\mathbf{n}_k^{\text{face}}$. A node $v_i \in V$ is surrounded by a set of faces forming a local disk-like structure, ensuring geometric continuity. The normal vector of a node $\mathbf{n}_i^{\text{node}}$ is defined as the average of the normal vectors of its surrounding faces: $\mathbf{n}_i^{\text{node}} = \frac{1}{|\mathcal{F}(v_i)|} \sum_{f_k \in \mathcal{F}(v_i)} \mathbf{n}_k^{\text{face}}$, where $\mathcal{F}(v_i)$ is the set of faces adjacent to node $v_i$, and $|\mathcal{F}(v_i)|$ is the number of such faces.

### 3.2. E(3)-Equivariance and Invariance

E(3)-equivariance ensures that a function's output transforms consistently under transformations from the Euclidean group E(3), which includes translations, rotations, and reflections. For a mapping $\phi : \mathcal{X} \to \mathcal{Y}$, $\phi$ is E(3)-equivariant if the following holds for all $g \in$ E(3):

$$\phi(T_g(x)) = T_g'(\phi(x)),$$

where $T_g : \mathcal{X} \to \mathcal{X}$ and $T_g' : \mathcal{Y} \to \mathcal{Y}$ are transformations applied to $x \in \mathcal{X}$ and $y \in \mathcal{Y}$, respectively.

*Invariance*, on the other hand, is a special case of equivariance where the output remains unchanged under transformations, satisfying: $\phi(T_g(x)) = \phi(x)$.

### 3.3. Thickness in the Mesh

Since traditional meshes lack explicit thickness information, we first define *thickness node pair* as a pair of nodes where one resides on one side of the surface and the other on the opposing side, aligned along the normal direction of the origin node. The thickness paired node of $v_i$ is denoted as $\mathcal{T}(v_i) \in V$, where $\mathcal{T}(v_i) \neq v_i$, and is defined as:

$$\mathcal{T}(v_i) = \underset{v_j \in V, v_j \neq v_i}{\arg \min} \|\mathbf{x}_j - (\mathbf{x}_i - d \cdot \mathbf{n}_i^{\text{node}})\|, \quad (1)$$

subject to $(\mathbf{x}_j - \mathbf{x}_i) \cdot \mathbf{n}_i^{\text{node}} < 0$, where $\mathbf{n}_i^{\text{node}}$ is the unit outward normal vector at node $v_i$, $d > 0$ is a scalar for the ray projection distance, and $\| \cdot \|$ is the Euclidean distance. This definition ensures that $\mathcal{T}(v_i)$ lies on the opposing side of the surface by requiring $(\mathbf{x}_j - \mathbf{x}_i) \cdot \mathbf{n}_i^{\text{node}} < 0$, avoiding the selection of nodes on the same surface.

The distance between thickness node pair is then defined as the *thickness* of the mesh at node $v_i$. Formally, the thickness $t(v_i)$ at node $v_i$ is given by:

$$t(v_i) = \|\mathbf{x}_i - \mathbf{x}_{\mathcal{T}(v_i)}\|. \quad (2)$$

This definition enables the quantification of mesh thickness at any node, providing a geometric measure of the spatial separation between opposing surfaces.

## 4. Methodology

Our method, T-EMNN, extends the encode-process-decode framework of MGN (Pfaff et al., 2020), introducing key innovations for handling 3D shapes with thickness while incorporating spatial information in an E(3)-equivariant manner. We employ a data-driven coordinate transformation (Sec. 4.1) to integrate spatial information. T-EMNN consists of an encoder (Sec. 4.2.1), a surface processor (Sec. 4.2.2), a thickness processor (Sec. 4.2.3), and a decoder (Sec. 4.2.4), effectively processing surfaces with thickness. The overall framework is shown in Fig. 2.

### 4.1. Coordinate Transformation: E(3)-Invariant Data-driven Coordinate System

Each node $v_i \in V$ has an original coordinate $\mathbf{x}_i^{\text{orig}}$, representing its position in the input coordinate system. To ensure E(3)-invariance, we transform the coordinates into a

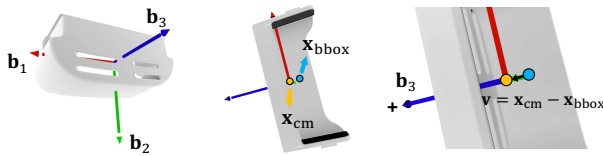

*Figure 2.* Overview of T-EMNN.

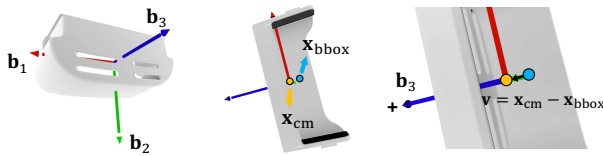

*Figure 3.* Our proposed data-driven coordinate system.

system defined by the shape itself, independent of its orientation or alignment in the original coordinate system. This transformation is achieved through the following steps:

**Step 1: Adjust Coordinates to Center of Mass.** The center of mass $\mathbf{x}_{\mathrm{cm}}$ of the shape is calculated as:

$$\mathbf{x}_{\mathrm{cm}} = \frac{1}{|V|} \sum_{v_i \in V} \mathbf{x}_i^{\mathrm{orig}}, \qquad (3)$$

where $|V|$ is the total number of nodes in the mesh. Each node's coordinate is adjusted relative to the center of mass:

$$\tilde{\mathbf{x}}_i = \mathbf{x}_i^{\mathrm{orig}} - \mathbf{x}_{\mathrm{cm}}. \qquad (4)$$

**Step 2. Principal Axis Generation.** As shown in Fig. 3, we obtain three orthogonal basis vectors $\mathbf{b}_1, \mathbf{b}_2, \mathbf{b}_3$, each in $\mathbb{R}^3$, by applying Principal Component Analysis (PCA) to $\tilde{\mathbf{X}} \in \mathbb{R}^{N \times 3}$, where $\tilde{\mathbf{X}}$ is composed of $\tilde{\mathbf{x}}$ defined in Eq. 4, and $N$ is the number of nodes within the mesh. These basis vectors form the principal axes of the shape and are combined into a rotation matrix $\mathbf{R} = \begin{bmatrix} \mathbf{b}_1 & \mathbf{b}_2 & \mathbf{b}_3 \end{bmatrix} \in \mathbb{R}^{3 \times 3}$.

**Step 3. Direction of Principal Axes.** To ensure consistent alignment of the principal axes, the direction of each basis vector $\mathbf{b}_{i \in \{1,2,3\}}$ is determined using a reference vector $\mathbf{v}$, which connects the center of the bounding box $\mathbf{x}_{\mathrm{bbox}}$ to the center of mass $\mathbf{x}_{\mathrm{cm}}$. The direction is adjusted as follows:

$$\mathbf{b}_i \leftarrow \begin{cases} \mathbf{b}_i, & \text{if } \mathbf{b}_i \cdot \mathbf{v} \geq 0, \\ -\mathbf{b}_i, & \text{if } \mathbf{b}_i \cdot \mathbf{v} < 0, \end{cases} \qquad (5)$$

where $\mathbf{b}_i \cdot \mathbf{v}$ determines whether the basis vector $\mathbf{b}_i$ aligns with the direction of $\mathbf{v}$, and $\mathbf{v} = \mathbf{x}_{\mathrm{cm}} - \mathbf{x}_{\mathrm{bbox}}$ is the reference vector. The center of the bounding box $\mathbf{x}_{\mathrm{bbox}}$ is computed as $\mathbf{x}_{\mathrm{bbox}} = \frac{\mathbf{x}_{\mathrm{min}} + \mathbf{x}_{\mathrm{max}}}{2}$, where

$$\mathbf{x}_{\mathrm{min}} = \begin{bmatrix} \min_{v \in V} x_v \\ \min_{v \in V} y_v \\ \min_{v \in V} z_v \end{bmatrix}, \quad \mathbf{x}_{\mathrm{max}} = \begin{bmatrix} \max_{v \in V} x_v \\ \max_{v \in V} y_v \\ \max_{v \in V} z_v \end{bmatrix}, \qquad (6)$$

are the minimum and maximum coordinates of the point cloud along each axis. This adjustment maintains consistent

principal axis orientation, regardless of the shape's initial alignment or transformations.

**Step 4. Coordinate Transformation.** The adjusted coordinates $\tilde{\mathbf{x}}_i$ are transformed into the E(3)-invariant coordinate system using the rotation matrix $\mathbf{R}$:

$$\mathbf{x}_i^{\mathrm{inv}} = \mathbf{R}^\top \tilde{\mathbf{x}}_i. \qquad (7)$$

**Preserving Original Coordinate Mapping.** For each shape, the center of mass $\mathbf{x}_{\mathrm{cm}}$ and the rotation matrix $\mathbf{R}$ are stored to enable the output predictions to be transformed back into the original coordinate system. During the inverse transformation, the predicted invariant coordinates $\mathbf{x}_i^{\mathrm{inv}}$ are mapped back to the original system as:

$$\mathbf{x}_i^{\mathrm{orig}} = \mathbf{R}\mathbf{x}_i^{\mathrm{inv}} + \mathbf{x}_{\mathrm{cm}}. \qquad (8)$$

This ensures that the predictions remain consistent with the original coordinate system while benefiting from the E(3)-invariant representation during processing. The transformed coordinates $\mathbf{x}_i^{\mathrm{inv}}$, along with the stored $\mathbf{x}_i$ and $\mathbf{R}$, allow seamless mapping between the input and output spaces.

We provide a formal proof of the invariance in Appendix H.

### 4.2. Thickness-aware Mesh Neural Network (T-EMNN)

#### 4.2.1. ENCODER

For every node $v_i \in V$ and edge $e_{ij} \in E$ within the surface mesh $M = (V, E)$, we encode their features using respective MLP encoders.

**Geometric Encoder.** To preserve E(3)-invariance during processing, T-EMNN utilizes only invariant features for geometric encoding (e.g., distance from the center mass or neighboring node) that remain robust under any transformation of the E(3) group. Formally, the encoded features for a node $v_i$ and an edge $e_{ij}$ are given as:

$$\mathbf{z}_i^{(0)} = \phi_{\mathrm{node}}(\mathbf{f}_i), \quad \mathbf{e}_{ij}^{(0)} = \phi_{\mathrm{edge}}(\mathbf{f}_{ij}), \qquad (9)$$

where $\mathbf{f}_i \in \mathbb{R}^{nf}$ and $\mathbf{f}_{ij} \in \mathbb{R}^{ef}$ represent the initial feature embeddings of node $v_i$ and edge $e_{ij}$. $\phi_{\mathrm{node}}$ and $\phi_{\mathrm{edge}}$ are MLP-based encoders designed for nodes and edges. The specific initial features utilized are detailed in Tab. 1.

The outputs of the geometric encoders, $\mathbf{z}_i^{(0)} \in \mathbb{R}^d$ and $\mathbf{e}_{ij}^{(0)} \in \mathbb{R}^d$, are later used as the input embeddings for the first layer ($l = 0$) of the processor modules.

**Spatial Encoder.** To incorporate spatial information (i.e., coordinates) into the analysis, T-EMNN utilizes an MLP-based spatial encoder to derive the spatial embedding $\mathbf{z}_i^{\text{coord}} \in \mathbb{R}^d$:

$$\mathbf{z}_i^{\text{coord}} = \phi_{\text{coord}}(\mathbf{x}_i^{\text{inv}}), \tag{10}$$

where $\mathbf{x}_i^{\text{inv}} \in \mathbb{R}^3$ represents the transformed coordinates of node $v_i$.

**Condition Encoder.** The condition embedding $\mathbf{h}_c \in \mathbb{R}^d$ is encoded as:

$$\mathbf{h}_c = \phi_{\text{cond}}(\mathbf{c}), \tag{11}$$

where $\phi_{\text{cond}}$ processes the experimental condition vector $\mathbf{c} \in \mathbb{R}^{cf}$, where $cf$ is the number of experimental conditions such as temperature and pressure.

Note that while the initial geometric embedding of node $\mathbf{z}_i^{(0)}$ and edge $\mathbf{e}_{ij}^{(0)}$ are passed through the processor, the spatial embedding $\mathbf{z}_i^{\text{coord}}$ and the condition embedding $\mathbf{h}_c$ are integrated with the processed geometric embeddings by the decoder after the processor. This allows the processor to focus on capturing the geometric features of the shape.

### 4.2.2. SURFACE PROCESSOR

The surface processor in T-EMNN focuses exclusively on the edges $e_{ij} \in E$ of the surface mesh $M$, without considering interactions between thickness node pairs. This processor updates the embeddings of the surface edges and nodes, capturing the geometric and topological relationships of the mesh surface. The update rule for the edge embeddings $\mathbf{e}_{ij}^{(l)} \in \mathbb{R}^d$ is defined as:

$$\mathbf{e}_{ij}^{(l+1)} \leftarrow f_{\text{surf}}^M(\mathbf{e}_{ij}^{(l)}, \mathbf{z}_i^{(l)}, \mathbf{z}_j^{(l)}), \tag{12}$$

where $f_{\text{surf}}^M$ is an MLP with residual connections, which updates the edge embeddings $\mathbf{e}_{ij}^{(l)}$ based on the embeddings of the connected nodes $v_i$ and $v_j$. Then, the update rule for the node embeddings $\mathbf{z}_i^{(l)} \in \mathbb{R}^d$ is defined as:

$$\mathbf{z}_i^{\text{surf},(l)} \leftarrow f_{\text{surf}}^V(\mathbf{z}_i^{(l)}, \sum_{j \in \mathcal{N}(i)} \mathbf{e}_{ij}^{(l+1)}), \tag{13}$$

where $f_{\text{surf}}^V$ is another MLP with residual connections and $\mathcal{N}(i)$ denotes the set of neighboring nodes of $v_i$. It updates the node embeddings $\mathbf{z}_i^{(l)}$ by aggregating the updated edge embeddings $\mathbf{e}_{ij}^{(l+1)}$ from its neighboring edges.

The updated node embeddings $\mathbf{z}_i^{\text{surf},(l)} \in \mathbb{R}^d$ serve as the input for the corresponding $l$-th layer of the thickness processor. This ensures that the geometric relationships refined by the surface processor are directly utilized by the thickness processor to model interactions between opposing surfaces.

### 4.2.3. THICKNESS PROCESSOR

The concept of thickness in meshes, and the methodology for identifying thickness node pair, has been detailed in

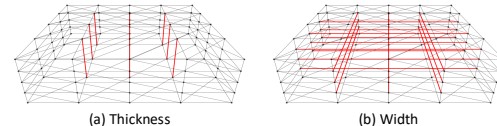

(a) Thickness         (b) Width

*Figure 4.* The concept of thickness (left) and width (right).

Sec 3.3. In brief, thickness is characterized by the spatial separation between opposing surfaces, with thickness paired node $\mathcal{T}(v_i)$ of node $v_i$ defined as the closest node on the opposing surface along the inward normal direction of a given node $v_i$. The thickness $t(v_i)$ defined in Eq. 2 represents the distance between opposing surfaces.

In addition, to account for thickness-related interactions, we introduce a thickness edge $e_{i,\text{thick}}$ connecting $v_i$ to $\mathcal{T}(v_i)$, with its feature $\mathbf{f}_{i,\text{thick}} \in \mathbb{R}^2$ defined as:

$$\mathbf{f}_{i,\text{thick}} = [t(v_i), \mathbf{n}_i \cdot \mathbf{n}_{i_\mathcal{T}}], \tag{14}$$

where $\mathbf{n}_i \cdot \mathbf{n}_{i_\mathcal{T}}$ is the dot product between the normal vectors at $v_i$ and $\mathcal{T}(v_i)$. Including $\mathbf{n}_i \cdot \mathbf{n}_{i_\mathcal{T}}$ ensures that the model accounts for the alignment of normal vectors between thickness pair nodes. This additional attribute complements the thickness $t(v_i)$, providing a more comprehensive representation of pairwise interactions for improved processing of geometric relationships.

**Thickness Threshold.** However, an important observation we had is that not all thickness node pair inherently represent the real-world meaning of thickness. For example, in the case of a wide flat plate (Fig. 4), the thickness node pairs on the side surfaces (Fig. 4(b)) are more representative of the width rather than the actual thickness of the object (Fig. 4(a)). The distinction between these two concepts is not discrete but rather ambiguous.

To address this, we define a *Thickness threshold* $\tau$ that dynamically regulates the interactions between nodes. Specifically, thickness edges are incorporated only for nodes whose distance to their thickness paired node is within the threshold ($t(v_i) \leq \tau$), ensuring meaningful interactions between opposing surfaces with strong dynamic relationships. It is important to note that the thickness threshold $\tau$ is not predefined but instead learned in a data-driven manner. By training on real-world data, the model dynamically adapts to identify the optimal threshold $\tau$ that captures interactions between opposing surfaces without relying on manual tuning or prior assumptions. This data-driven approach enhances the framework's robustness and adaptability across various applications.

**Thickness Activation Function.** The thickness activation value $I_i$ is calculated for each node $v_i$ to determine the contribution of its thickness edge in the propagation process. The activation is defined as:

$$I_i = \frac{1}{1 + e^{\alpha(t(v_i) - \tau)}} \tag{15}$$

where $t(v_i)$ represents the thickness of node $v_i$, $\tau$ is a learnable threshold, and $\alpha$ is a scaling factor set to 3, controlling transition sharpness. By doing so, edges with $t(v_i) \leq \tau$ are assigned weights close to 1, while edges with $t(v_i) > \tau$ are excluded from the propagation process.

**Weighted Message Passing.** In the thickness processor, each node $v_i$ is connected to its thickness paired node $\mathcal{T}(v_i)$ through a single thickness edge $e_{i,\text{thick}}$. The embedding for this thickness edge $\mathbf{e}_{i,\text{thick}} \in \mathbb{R}^d$ is initialized in the first layer using a dedicated encoder, $\phi_{\text{thick}}$, which maps the thickness edge feature $\mathbf{f}_{i,\text{thick}}$ to its embedding:

$$\mathbf{e}_{i,\text{thick}}^{(0)} \leftarrow \phi_{\text{thick}}(\mathbf{f}_{i,\text{thick}}), \tag{16}$$

where $\mathbf{f}_{i,\text{thick}}$ is defined in Eq. 14. For subsequent layers, $\mathbf{e}_{i,\text{thick}}^{(l)} \in \mathbb{R}^d$ is updated using the output from the previous layer $\mathbf{e}_{i,\text{thick}}^{(l-1)}$. This ensures effective propagation of thickness and normal alignment information through the layers.

The updated embedding for edge $\mathbf{e}_{i,\text{thick}}^{(l+1)}$ at each layer is computed as:

$$\mathbf{e}_{i,\text{thick}}^{(l+1)} \leftarrow I_i \cdot f_{\text{thick}}^M(\mathbf{e}_{i,\text{thick}}^{(l)}, \mathbf{z}_i^{\text{surf},(l)}, \mathbf{z}_{\mathcal{T}(v_i)}^{\text{surf},(l)}), \tag{17}$$

where $f_{\text{thick}}^M$ is an MLP that processes the features of the thickness edge $e_{i,\text{thick}}$, as well as the node embeddings $\mathbf{z}_i^{\text{surf},(l)}$ and $\mathbf{z}_{\mathcal{T}(v_i)}^{\text{surf},(l)}$. The activation value $I_i$ ensures that only relevant thickness interactions contribute during message propagation. Then, the updated embedding of node $v_i$, $\mathbf{z}_i^{(l+1)} \in \mathbb{R}^d$, is then computed as:

$$\mathbf{z}_i^{(l+1)} \leftarrow f_{\text{thick}}^V(\mathbf{z}_i^{\text{surf},(l)}, \mathbf{e}_{i,\text{thick}}^{(l+1)}), \tag{18}$$

where $f_{\text{thick}}^V$ is another MLP that combines the current node embedding $\mathbf{z}_i^{\text{surf},(l)}$ with the updated edge embedding $\mathbf{e}_{i,\text{thick}}^{(l+1)}$.

By focusing on a single thickness edge per node, this message-passing scheme maintains computational efficiency while retaining the ability to model relationships between opposing surfaces. The architecture of the thickness processor ensures seamless integration of both surface-level and thickness-based geometric information.

### 4.2.4. DECODER

In the decoder, T-EMNN produces the final node-level predictions by combining the geometric embedding $\mathbf{z}_i$, spatial embedding $\mathbf{z}_i^{\text{coord}}$, and experimental conditions $\mathbf{h}_c$.

First, the geometric and spatial embeddings are concatenated and processed as follows:

$$\mathbf{z}_i^{\text{final}} = \phi_{\text{combine}}([\mathbf{z}_i, \mathbf{z}_i^{\text{coord}}]) \in \mathbb{R}^d, \tag{19}$$

where $\phi_{\text{combine}}$ integrates geometric and spatial features. Then, the combined embedding $\mathbf{z}_i^{\text{final}}$ is concatenated with $\mathbf{h}_c$ and passed through the decoder:

$$\mathbf{p}_i^{\text{inv}} = \phi_{\text{decode}}([\mathbf{z}_i^{\text{final}}, \mathbf{h}_c]), \tag{20}$$

producing the node-level prediction $\mathbf{p}_i^{\text{inv}} \in \mathbb{R}^3$ in the transformed E(3)-invariant data-driven coordinate system.

**Inverse Transformation.** To map the prediction back to the original coordinate system, the stored center of mass $\mathbf{x}_{\text{cm}}$ and rotation matrix $\mathbf{R}$ for the shape are used. The final deformation in the original system is calculated as:

$$\mathbf{p}_i^{\text{orig}} = \mathbf{R} \cdot \mathbf{p}_i^{\text{inv}} + \mathbf{x}_{\text{cm}}.$$

This ensures that the predicted deformations are consistent with the input shape's original orientation and alignment, while benefiting from the E(3)-invariant processing during model computation.

## 5. Experiment

### 5.1. Dataset Description

We evaluate T-EMNN using a dataset from real-world injection molding applications. Injection molding, a common manufacturing process, involves injecting molten plastic or metal into a mold, letting it solidify, and then taking the finished product out of the mold. Product quality depends on factors like temperature, geometry, and gate design, with deformation being critical. This dataset is well-suited for evaluating T-EMNN as its geometries exhibit thickness across all surfaces, enabling thickness-related interaction modeling. Additionally, node spatial positions significantly influence deformation, underscoring the importance of geometric and spatial information. The predominantly "basket-like" structures capture both surface-level and thickness-based interactions well. More details, including data split and initial features, are provided in the Appendix A.

### 5.2. Baselines

As baselines, we include multiple graph-based neural network methods to evaluate T-EMNN against existing techniques. **MGN** (Pfaff et al., 2020) models physical interactions using graph-based message passing but lacks E(3)-equivariance. **EGNN** (Satorras et al., 2021) ensures E(3)-equivariance through message passing while maintaining computational efficiency by avoiding direct nonlinear encoding of spatial features. Building upon EGNN, **EMNN** (Trang et al., 2024) optimizes this framework for mesh data by generating E(3)-invariant messages that incorporate geometric information from mesh faces.

### 5.3. Evaluation Settings

We assess the models under two distinct conditions: 1) *in-distribution*, where the test data retains the same aligned coordinate system as the training data, and 2) *out-of-distribution*, where the original coordinate system is randomly rotated, resulting in a misalignment with the training data. Note that the out-of-distribution scenario is designed to assess how well the methods adapt to objects

*Table 1.* Model Performance in In-Distribution and Out-of-Distribution Settings, averaged over 3 seeds with standard deviation (in parentheses). **Bold** indicates the best performance among the methods. Descriptions of each feature are provided in Appendix A.

| Model | Equivariance | Spatial information | Thickness edges | Input of $\phi_{coord}$ | Edge Feature $f_{ij}$ | Node Feature $f_i$ | In Distribution (Original) | | | Out of Distribution (Rotated) | | |
|---|---|---|---|---|---|---|---|---|---|---|---|---|
| | | | | | | | RMSE ($\downarrow$) | MAE ($\downarrow$) | $R^2$ ($\uparrow$) | RMSE ($\downarrow$) | MAE ($\downarrow$) | $R^2$ ($\uparrow$) |
| (a) MLP | $\times$ | $\checkmark$ | $\times$ | - | - | $x^{orig}$ | 0.2818 (0.0061) | 0.1164 (0.0035) | 0.8984 (0.0029) | 0.4789 (0.0181) | 0.1939 (0.0070) | 0.7393 (0.0248) |
| (b) MLP | $\checkmark$ | $\checkmark$ | $\times$ | - | - | $x^{inv}$ | 0.2546 (0.0015) | 0.1043 (0.0008) | 0.9154 (0.0016) | 0.2545 (0.0015) | 0.1071 (0.0007) | 0.9385 (0.0009) |
| (c) MGN | $\times$ | $\times$ | $\times$ | - | $x_{ij}, \|x_{ij}\|$ | $n_i, g_i, r_i$ | 1.2608 (0.0107) | 0.5607 (0.0041) | 0.0782 (0.0315) | 1.3188 (0.0164) | 0.6199 (0.0064) | -0.0903 (0.0315) |
| (d) MGN | $\times$ | $\checkmark$ | $\times$ | $x^{orig}$ | $x_{ij}, \|x_{ij}\|$ | $n_i, g_i, r_i$ | 0.2854 (0.0046) | 0.1176 (0.0017) | 0.8724 (0.0037) | 0.4514 (0.0190) | 0.1938 (0.0067) | 0.7917 (0.0180) |
| (e) MGN | $\checkmark$ | $\checkmark$ | $\times$ | $x^{inv}$ | $x_{ij}, \|x_{ij}\|$ | $n_i, g_i, r_i$ | 0.2241 (0.0042) | 0.0938 (0.0029) | 0.9113 (0.0099) | 0.2241 (0.0042) | 0.0965 (0.0024) | 0.9446 (0.0033) |
| (f) EGNN | $\checkmark$ | $\times$ | $\times$ | - | $\|x_{ij}\|$ | $g_i, r_i$ | 153.051 (4.2992) | 54.363 (2.1000) | -14341.0 (1214.1) | 196.343 (1.6422) | 89.049 (1.2804) | -32260.9 (1039.3) |
| (g) EGNN | $\times$ | $\checkmark$ | $\times$ | $x^{orig}$ | $\|x_{ij}\|$ | $g_i, r_i$ | 0.2944 (0.0045) | 0.1220 (0.0021) | 0.8680 (0.0056) | 0.4576 (0.0184) | 0.1958 (0.0064) | 0.8074 (0.0206) |
| (h) EGNN | $\checkmark$ | $\checkmark$ | $\times$ | $x^{inv}$ | $\|x_{ij}\|$ | $g_i, r_i$ | 0.2270 (0.0019) | 0.0963 (0.0008) | 0.9129 (0.0026) | 0.2271 (0.0019) | 0.0987 (0.0009) | 0.9443 (0.0012) |
| (i) EMNN | $\checkmark$ | $\times$ | $\times$ | - | $\|x_{ij}\|$ | $g_i, r_i$ | 166.077 (1.5226) | 58.467 (2.0000) | -16034.0 (975.8) | 201.450 (1.7433) | 92.237 (1.3366) | -34302.7 (644.62) |
| (j) EMNN | $\times$ | $\checkmark$ | $\times$ | $x^{orig}$ | $\|x_{ij}\|$ | $g_i, r_i$ | 0.3056 (0.0246) | 0.1284 (0.0131) | 0.8626 (0.0052) | 0.4668 (0.0180) | 0.2024 (0.0092) | 0.7972 (0.0097) |
| (k) EMNN | $\checkmark$ | $\checkmark$ | $\times$ | $x^{inv}$ | $\|x_{ij}\|$ | $g_i, r_i$ | 0.2210 (0.0057) | 0.0937 (0.0034) | 0.9149 (0.0034) | 0.2210 (0.0057) | 0.0963 (0.0052) | 0.9473 (0.0012) |
| (l) T-EMNN | $\checkmark$ | $\checkmark$ | $\checkmark$ | $x^{inv}$ | $\|x_{ij}\|$ | $g_i, r_i$ | **0.2132** (0.0046) | **0.0892** (0.0025) | **0.9228** (0.0063) | **0.2131** (0.0046) | **0.0918** (0.0023) | **0.9513** (0.0031) |

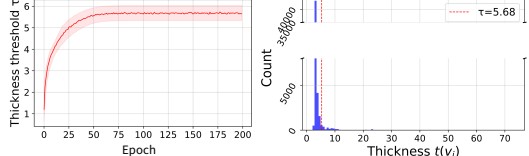

*Figure 5.* Learning curve of the thickness threshold $\tau$ during training across three seeds (left), and the distribution of thickness values $t(v_i)$ with the cutoff threshold (red dotted line, $t(v_i) = \tau$) used for message passing in the thickness processor (right).

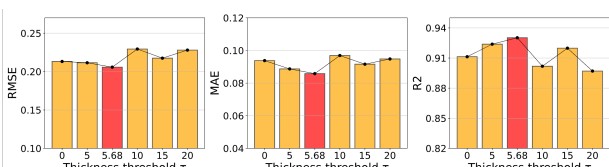

*Figure 6.* Performance comparison of T-EMNN with a fixed thickness threshold. The value 5.68 corresponds to the learned thickness threshold in T-EMNN when using a learnable threshold.

with E(3)-transformed coordinates, where maintaining E(3)-equivariance becomes essential.

We assess the model performance using three metrics: 1) RMSE, which evaluates the effectiveness of handling outliers, 2) MAE, which measures the consistency and accuracy of the model's predictions, and 3) $R^2$, which quantifies the model's ability to explain variance, ensuring reliability for real-world industrial applications.

### 5.4. Experiment Results

To evaluate data-driven coordinates, we modify the baselines to use spatial embeddings (i.e., $z_i^{coord}$) derived from either the original ($x_i^{orig}$) or proposed ($x_i^{inv}$) coordinates (Sec. 5.4.1). Additionally, we assess whether the learned thickness threshold $\tau$ optimally facilitates message passing between opposing surfaces (Sec. 5.4.2).

#### 5.4.1. MAIN RESULTS

In Tab. 1, we observe T-EMNN, which incorporates spatial information into the model while enabling propagation between opposing surfaces with strong relationships, outperforms other methods. The detailed analysis is as follows:

**Impact of Spatial Information.** Spatial information plays a critical role in capturing localized patterns, which are essential for accurate interpretation in downstream tasks. To evaluate the isolated impact of spatial information, we first evaluate the performance of MLPs that use only vertex coordinates as input ((a) and (b) in Tab. 1). The results demonstrate that spatial information alone is sufficient to achieve strong performance in terms of $R^2$ score, highlighting its importance in representing meaningful relationships and

patterns in the data. However, when the coordinate system lacks E(3)-equivariant properties, performance significantly deteriorates when testing data exhibits a different coordinate distribution (i.e., out-of-distribution results of (a) in Tab. 1). This underscores the critical role of E(3)-equivariance in ensuring the robustness of the coordinate system.

Moreover, for baselines that do not explicitly leverage latent spatial embeddings (e.g., (c) MGN, (f) EGNN, and (i) EMNN in Tab. 1), they solely rely on geometric relationships between neighboring edges or faces within the mesh graph. However, when observing $R^2$ values, relying solely on local geometric relations (e.g., coordinate differences, distances) proves insufficient for accurately predicting regional behavior across the entire shape.

**Impact of Data-driven Coordinate.** To address the missing spatial information in baseline methods, we incorporate an additional spatial encoder, as defined in Eq. 10, and combine it with the geometric embedding via concatenation, as described in Eq. 19. When integrating our proposed data-driven coordinate system into the baselines, an inverse transformation of the outputs is applied to ensure E(3)-equivariance.

In Tab. 1, methods utilizing our data-driven coordinates ((e) MGN, (h) EGNN, and (k) EMNN) consistently outperform their counterparts using the original coordinate system ((d) MGN, (g) EGNN, and (j) EMNN). The enhanced alignment provided by our proposed data-driven coordinate system significantly improves the representation of spatial relationships, leading to superior performance in downstream tasks. Furthermore, the data-driven coordinate system achieves this not only by preserving E(3)-equivariance ((h) EGNN and (k) EMNN) but also by enabling E(3)-equivariance in

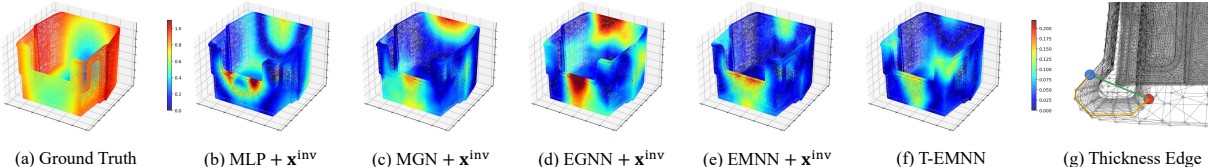

(a) Ground Truth     (b) MLP + $\mathbf{x}^{\text{inv}}$     (c) MGN + $\mathbf{x}^{\text{inv}}$     (d) EGNN + $\mathbf{x}^{\text{inv}}$     (e) EMNN + $\mathbf{x}^{\text{inv}}$     (f) T-EMNN     (g) Thickness Edge

*Figure 7.* Visualization of error magnitude (RMSE). The ground truth shows deformation magnitude (a), while (b–f) illustrate prediction errors. Additional examples are in Fig. 15 (Appendix). In (g), (●) and (●) represent a thickness node pair, (—) its thickness edge, and (—) the shortest path within the mesh of length 6.

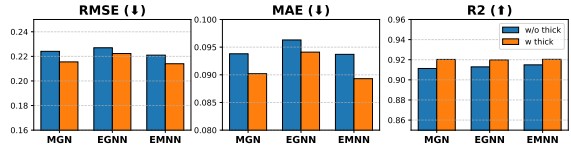

*Figure 8.* Comparison of baselines with $\mathbf{x}^{\text{inv}}$ and their extension with our thickness edges.

models that originally lacked it, such as (e) MGN. This capability transforms previously limited models into robust systems capable of handling transformations effectively, thereby greatly enhancing their ability to capture and represent spatial relationships.

### 5.4.2. IMPACT OF THICKNESS-AWARE FRAMEWORK

To assess the effectiveness of the thickness-aware framework, we examine whether the learned thickness threshold $\tau$ optimizes message passing between opposing surfaces and distinguishes 'thickness' from 'width.' As shown in Fig. 5, $\tau$ converges to 5.68 across three seeds with low variance, filtering out 3.83% of thickness edges exceeding this threshold (Sec. 3.3). In this section, we verify whether the thickness threshold learned by our model impacts performance and accurately captures the actual thickness.

**Performance Impact of Thickness Edges.** To confirm that $\tau = 5.68$ is optimal, we compare the performance of T-EMNN using fixed thresholds ranging from 0 to 20 (Fig. 6). The results show that performance peaks when the threshold is near 5.68. Moreover, when the threshold is set to zero—removing thickness edges—performance degrades significantly. Similarly, thresholds above 10 introduce noisy information from irrelevant nodes representing 'width,' leading to performance deterioration.

To further examine the importance of thickness edges, we incorporate them, along with our thickness processor, into the baseline models. As shown in Fig. 8, all baseline models exhibit improved performance when incorporating thickness edges compared to their counterparts without them. This result validates the importance of the thickness in 3D objects, and its effective integration can improve the models' capability in static analysis.

**Ablation Study of Thickness Edge Features.** In Tab. 2, we analyze the performance impact of the proposed thickness edge features, $\mathbf{f}_{i,\text{thick}}$, comprising two components: the *thick-*

*Table 2.* Ablation study on the thickness edge feature, $\mathbf{f}_{i,\text{thick}}$.

| Method | RMSE | MAE | $\mathbf{R}^2$ |
|---|---|---|---|
| w/o thickness | 0.2156 (0.0064) | 0.0908 (0.0027) | 0.9148 (0.0089) |
| w/o dot product | 0.2191 (0.0143) | 0.0912 (0.0067) | 0.9134 (0.0163) |
| T-EMNN | **0.2132** (0.0046) | **0.0892** (0.0025) | **0.9228** (0.0063) |

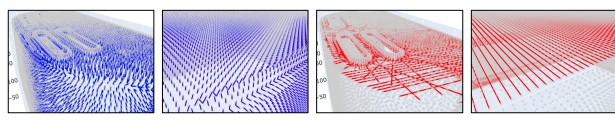

(a) Below the threshold $\tau$       (b) Above the threshold $\tau$

*Figure 9.* Visualization of thickness edges with $t(v_i)$ below the learned threshold $\tau$ (left) and those above $\tau$, filtered out by the thickness processor (right). Detailed in Fig. 16 (Appendix).

*ness* $t(v_i)$ and the *dot product* of normal vectors $\mathbf{n}_i \cdot \mathbf{n}_{i_\tau}$. The thickness feature encodes the weight of the edges based on the distance between thickness pairs, while the dot product of normal vectors imposes geometric alignment by quantifying the directional consistency between the normals of the paired nodes. By jointly leveraging these features, the model effectively learns edge weights that promote high-quality message passing between opposing surfaces, thereby enhancing overall performance.

**Qualitative Analysis.** Fig. 9 illustrates that the learned thickness threshold effectively filters out noisy thickness edges while retaining meaningful ones, thereby enhancing the model's ability to process relevant interactions.

In Fig. 7 (b-e), we observe baselines struggle to predict around edges, where stress concentration leads to distinct physical behavior. This is because the inherent locality of surface-mesh, where it requires GNN-based methods to take at least six propagation steps along the shortest path (— in Fig. 7(g)). In contrast, T-EMNN facilitates single-hop interactions between opposing surface neighbors through thickness edges (— in Fig. 7(g)), allowing for broader message passing and improving performance in these crucial regions in Fig. 7(f).

### 5.4.3. EVALUATION UNDER DYNAMIC SETTING

To evaluate the dynamic capabilities of our framework—particularly the thickness processor—we conduct next-timestep deformation prediction using the Deforming Plate dataset (Pfaff et al., 2020). This experiment demonstrates how the thickness processor enhances the model's ability to handle dynamic scenarios.

*Table 3.* Performance evaluation on the **Deforming Plate dataset** for predicting deformation at the next timestep.

| Method | Input of $\phi_{\mathbf{coord}}$ | RMSE ($\times 10^3$) | MAE ($\times 10^3$) | $\mathbf{R}^2$ |
|---|---|---|---|---|
| **T-EMNN** w/o thickness | $\mathbf{x}^{\mathrm{orig}}$ | 17.420 (3.022) | 6.830 (1.039) | 0.7007 (0.0524) |
| **T-EMNN** | $\mathbf{x}^{\mathrm{orig}}$ | **14.903** (0.671) | **5.851** (0.195) | **0.7579** (0.0220) |

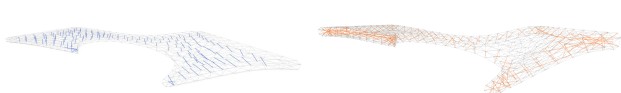

*Figure 10.* Visualization of thickness edges in the **Deforming Plate dataset**: edges with $t(v_i)$ below the learned threshold $\tau$ (left), and edges with $t(v_i)$ above $\tau$, which are filtered out by the thickness processor (right).

We first construct thickness edges based on the mesh at the initial timestep and keep them fixed throughout all subsequent timesteps. The prediction task is to estimate the deformation of each node at the next timestep. As our data-driven coordinate system is designed for static analysis and to mitigate misalignment within the dataset, we instead adopt the original coordinate system for dynamic analysis. Furthermore, because our method assumes a single material, we incorporate an additional node feature: the shortest distance from each node to the actuator in the global coordinate system. This allows the model to capture external interactions applied by the actuator on the deforming plate.

As shown in Tab. 3, integrating the thickness edges constructed by our thickness processor leads to better performance by accounting for the real-world thickness of the plate. Fig. 10 illustrates that our thickness processor effectively forms edges between nodes that represent true material thickness (left) while effectively removing noisy connections that do not conform to the thickness criteria defined by our framework (right).

## 6. Conclusion

We introduced T-EMNN, a thickness-aware E(3)-equivariant 3D mesh neural network that captures complex geometric interactions, including thickness, by integrating a learnable thickness threshold for effective message passing while filtering irrelevant connections. By leveraging a data-driven coordinate system and a transformation mechanism that preserves E(3)-equivariance, T-EMNN achieves state-of-the-art performance in predicting node-level 3D deflection with high accuracy and computational efficiency. Comprehensive evaluations on real-world industrial datasets validate its robustness and practicality, making it an effective solution for applications like injection molding.

## Acknowledgements

This work was supported by LG Electronics; by the National Research Foundation of Korea (NRF) grant funded by the Korea government (MSIT) (RS-2024-00335098); and by the National Research Foundation of Korea (NRF) funded by Ministry of Science and ICT (RS-2022-NR068758).

## Impact Statement

This paper presents work whose goal is to advance the field of Machine Learning. There are no immediate ethical concerns associated with this work. However, as with any advancement in machine learning, its application in real-world scenarios should be carefully evaluated to prevent potential biases or unintended consequences.

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

## A. Data Details

The dataset consists of 504 valid samples derived from 28 unique geometries, each with 18 experimental conditions (i.e., $28 \times 18 = 504$). The training and validation sets include 23 geometries, with 80% of the samples used for training and the rest for validation. One geometry from the training set was entirely reserved for validation. The test set comprises all 18 experimental conditions for 5 geometries and the 18th experimental condition for the remaining 23 geometries, resulting in 113 test samples. The average mesh consists of approximately 54,127 nodes and 324,771 edges.

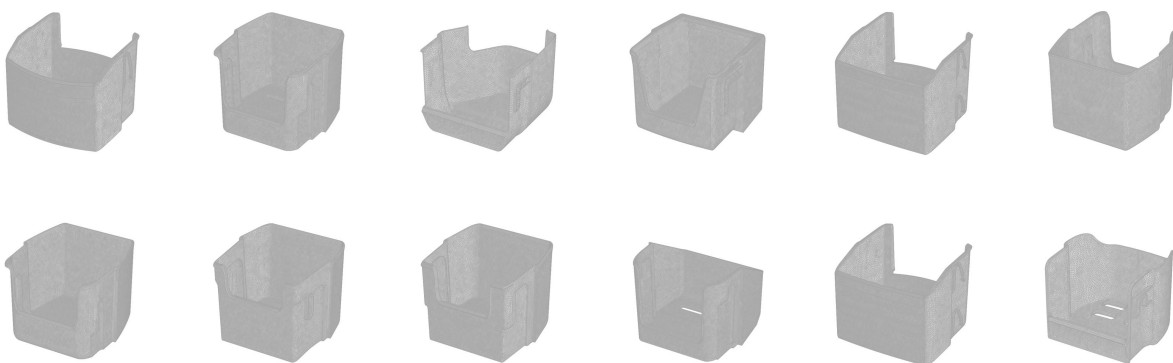

*Figure 11.* Examples of Dataset Shapes.

The experimental conditions consist of eight types for each shape: pack pressure, pack time, projected area, gate size, injection value, volume, melt temperature, and mold temperature. By varying these combinations, a total of 18 experimental conditions are generated.

MGN utilizes the coordinate difference between nodes $i$ and $j$ (i.e., $\mathbf{x}_{ij}$) and their distance (i.e., $\|\mathbf{x}_{ij}\|$) as edge features. For node features, MGN incorporates the normal vector of node $i$ (i.e., $\mathbf{n}_i$), the geodesic distance from the gate (the injection position of molten plastic or metal) to node $i$ (i.e., $g_i$), and the radius from the center of mass to node $i$ (i.e., $r_i$). EGNN, EMNN and T-EMNN, which ensure E(3)-equivariance, use only $\|\mathbf{x}_{ij}\|$ as edge features and $g_i$ and $r_i$ as node features.

## B. Implementation and Experimental Setup

Our model is implemented using Python 3.10.13, PyTorch 2.0.1, Torch-Geometric 2.4.0, and trimesh 3.23.5. All experiments were conducted on an NVIDIA GeForce RTX 4090 with CUDA 12.2.

Each experiment was run for 200 epochs per seed with a learning rate of 0.001 and a weight decay of 5e-4. To ensure stable optimization of the learnable thickness threshold $\tau$, we employ an adaptive learning rate scheduling strategy. Specifically, we utilize the ReduceLROnPlateau algorithm, which dynamically adjusts the learning rate when a monitored metric plateaus. We set the patience to 5, the initial threshold to 1, and the reduction factor to 0.5, applying it exclusively to the learnable thickness threshold.

## C. Baseline Details

In this work, we compare three prominent graph-based methods. MGN (Pfaff et al., 2020) utilizes iterative message-passing steps, while EGNN (Satorras et al., 2021) and EMNN (Trang et al., 2024) are designed for E(3)-equivariance. To ensure consistency, we rely on the official codes provided by the original authors for all baselines.

- **MGN:** https://github.com/google-deepmind/deepmind-research/ tree/master/meshgraphnets

- **EGNN:** https://github.com/vgsatorras/egnn

- **EMNN:** https://github.com/HySonLab/EquiMesh

To ensure a fair comparison, we configure all baselines and our proposed method, T-EMNN, with three message-passing

layers and 32 hidden dimensions. All results are reported based on the performance of the best validation model selected within 200 epochs across 3 different seeds. The spatial and condition encoders consist of two linear layers with ReLU activations. Since our task involves static analysis without time-step predictions, all baselines directly predict the target field variables, specifically the 3D deformation.

## D. Hyperparameter Details

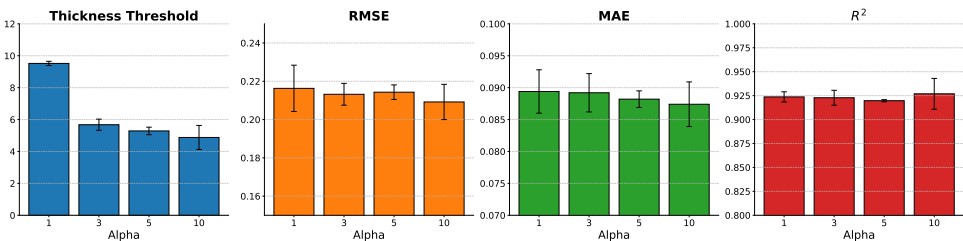

*Figure 12.* Analysis of Hyperparameter $\alpha$.

Hyperparameter $\alpha$ is a scaling factor that influences the transition sharpness. Specifically, as $\alpha$ increases, the thickness activation value $I_i$ in Eq. 15 discretely masks the thickness edges whose thickness values are near the threshold. In contrast, when $\alpha$ is small, $I_i$ smoothly attenuates the weight of thickness edges, gradually approaching zero beyond the threshold.

A smaller $\alpha$ ensures more stable learning of the thickness threshold by providing a smoother transition. However, it may introduce noise from softly masked thickness edges, potentially degrading the overall model performance. As shown in Fig. 12, when $\alpha$ is between 3 and 10, the thickness threshold converges within the range of 4 to 6. Conversely, when $\alpha = 1$, longer edges contribute to message passing, leading to less precise control over edge masking. When $\alpha$ is large (e.g., $\alpha = 10$), edges around the thickness threshold are strictly masked, but the convergence becomes unstable due to discrete masking effects, resulting in fluctuations.

Based on these observations, we set $\alpha$ to 3, which provides stable convergence for learning the thickness threshold while maintaining acceptable performance.

## E. Computational Efficiency

| Model | Input of $\phi_{\text{coord}}$ | Speed (it/s) | GPU Memory (MB) |
|---|---|---|---|
| MLP | - | 32.51 | 693 |
| MLP | - | 32.72 | 693 |
| MGN | - | 21.18 | 1,656 |
| MGN | $\mathbf{x}^{\text{orig}}$ | 22.58 | 3,954 |
| MGN | $\mathbf{x}^{\text{inv}}$ | 22.29 | 3,952 |
| EGNN | - | 23.85 | 5,700 |
| EGNN | $\mathbf{x}^{\text{orig}}$ | 23.59 | 5,740 |
| EGNN | $\mathbf{x}^{\text{inv}}$ | 23.66 | 5,740 |
| EMNN | - | 18.99 | 7,250 |
| EMNN | $\mathbf{x}^{\text{orig}}$ | 19.75 | 7,488 |
| EMNN | $\mathbf{x}^{\text{inv}}$ | 19.99 | 7,322 |
| T-EMNN | $\mathbf{x}^{\text{inv}}$ | 20.21 | 3,714 |

*Table 4.* Comparison of training speed (iteration/sec) and GPU memory usage (MB) across different models.

Our model is based on MGN, with an additional thickness edges module that introduces minimal computational overhead. This ensures that the computational cost remains similar to MGN while improving structural representation.

**Training Speed (it/s).** T-EMNN achieves **20.21 it/s**, comparable to MGN and slightly higher than EMNN (**18.99–19.99 it/s**). This indicates that the added module does not significantly impact training speed.

**GPU Memory Usage.** T-EMNN consumes **3,714 MB** of GPU memory, much lower than EMNN (7,250–7,488 MB) and similar to MGN. This shows that our approach maintains efficiency while reducing memory overhead.

## F. Result Details

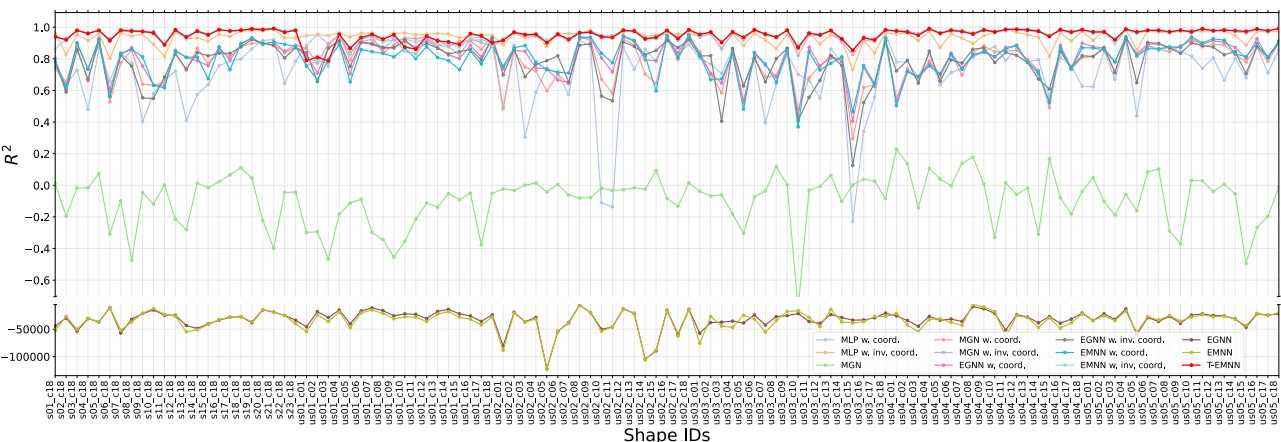

Figure 13. $R^2$ scores for all test data. In the shape IDs, 's' indicates seen shapes included in the training data, while 'us' refers to unseen shapes. The number following 's' or 'us' represents the shape type, and 'c' denotes the experimental condition, followed by its corresponding type (e.g., 's02_c18').

In Figure 13, we present the detailed results for each test dataset under Out-of-Distribution (OOD) settings. While GNN-based methods incorporating our proposed data-driven coordinate system (e.g., MGN w/ inv. coord., EGNN w. inv. coord., EMNN w. inv. coord., and T-EMNN) generally exhibit superior performance across all shapes, methods that utilize the original coordinate system (e.g., MGN w. coord., EGNN w. coord., and EMNN w. coord.) perform significantly worse. Furthermore, MGN, EGNN, and EMNN, which rely solely on deep representations of geometric features, fail to accurately predict the target deformation.

## G. Surface Mesh vs. Volume Mesh

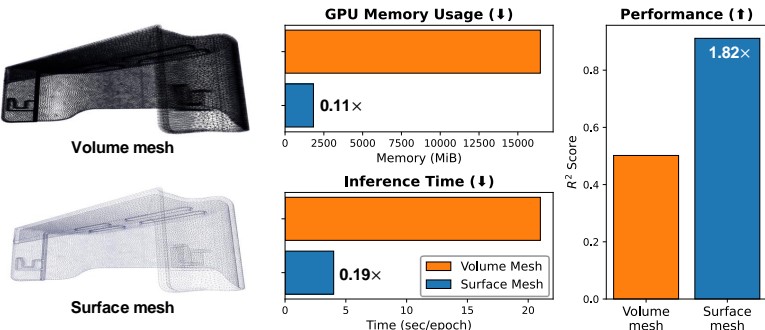

Figure 14. Comparisons between volume mesh and surface mesh. The methods used for comparison are based on the MGN framework with coordinate embeddings from our proposed coordinate system. GPU memory usage represents the average GPU consumption across the test dataset, and inference time reflects the total time required to process the test dataset.

Mesh representations can be categorized into surface and volume meshes, each with unique strengths and limitations. Surface meshes, representing only the outer boundary of 3D objects, are computationally efficient and excel at capturing geometric and topological properties. However, they cannot model internal structures, which are critical for analyzing physical interactions within the object.

Volume meshes, in contrast, extend to the interior of objects, enabling high-fidelity analyses such as FEM for properties like density, thermal gradients, and stress. However, as shown in Figure 14, they are computationally expensive and require significant effort to preprocess CAD models into high-quality meshes. Additionally, their dense internal connectivity poses challenges for tasks focused on understanding geometry and topology. In GNN-based methods, nodes in volume meshes receive messages from all surrounding nodes, which are processed based on relative distances and directions. This excessive connectivity complicates the network's ability to clearly capture the overall shape and structure, ultimately hindering performance in geometry-driven tasks.

Surface meshes, with their simplified representation of outer boundaries, are better suited for tasks prioritizing geometry and topology. However, they lack edges that connect opposing surfaces to represent thickness and capture critical correlations. This limitation challenges GNNs to effectively model interactions between these surfaces, often necessitating extended message-passing steps. Such methods can lead to issues like over-smoothing (Li et al., 2018a) or over-squashing (Alon & Yahav, 2020), limiting their overall effectiveness.

To overcome these limitations, we propose the Thickness-Aware Mesh Neural Network (T-EMNN), which combines the computational efficiency of surface meshes with enhanced modeling of internal interactions. By incorporating a thickness-aware message-passing mechanism, our method captures correlations between opposing surfaces while preserving the geometric and topological strengths of surface meshes. This enables robust and efficient analysis of 3D objects with intrinsic thickness, effectively bridging the gap between surface and volume meshes.

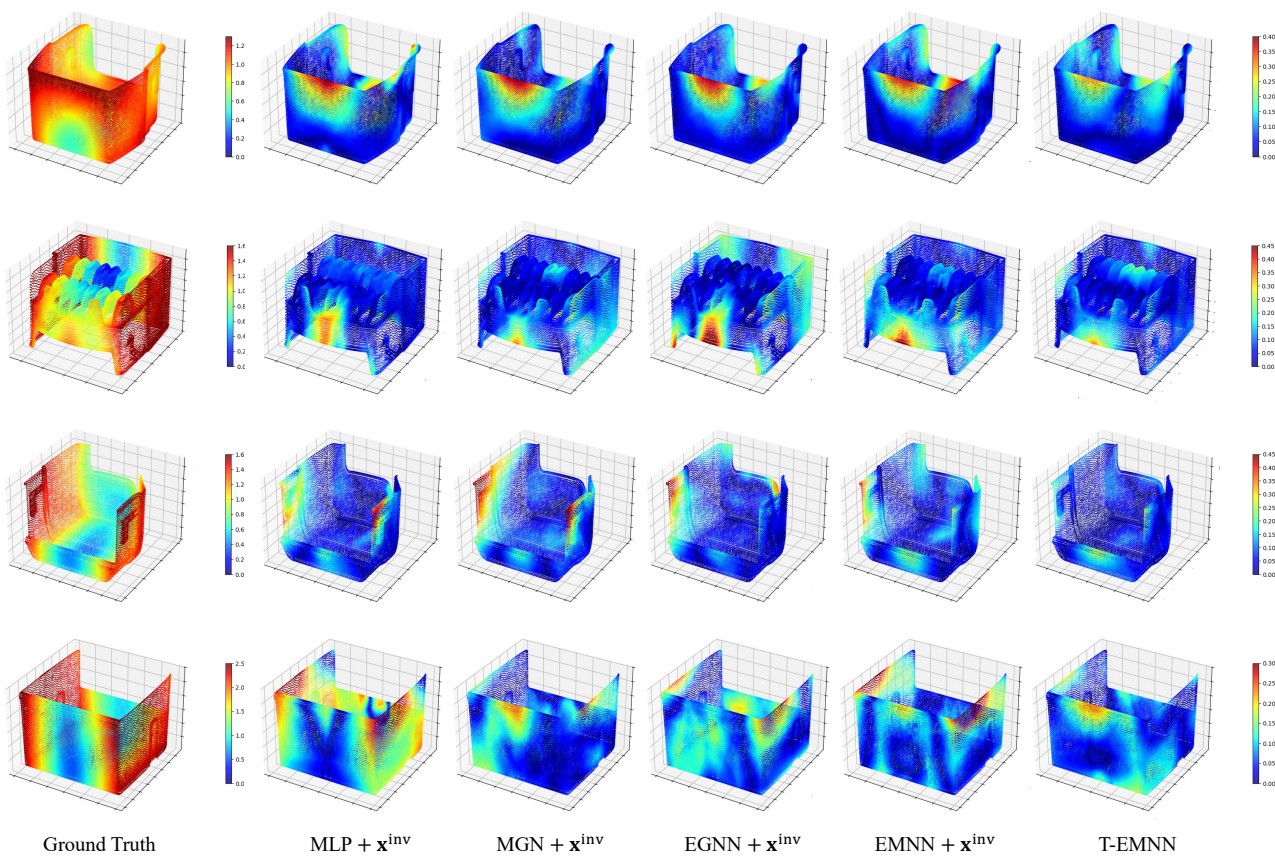

*Figure 15.* Visualization of error magnitude (RMSE). The ground truth represents the magnitude of deformation, while each method's figure illustrates the prediction error (RMSE) relative to the ground truth.

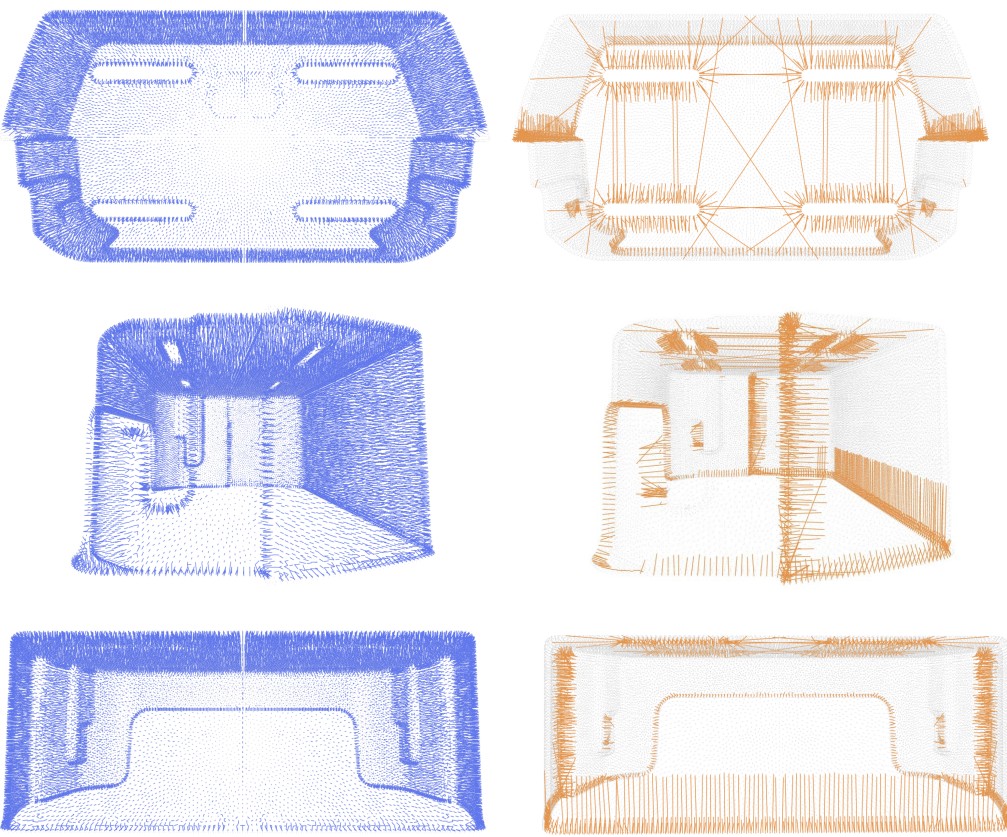

*Figure 16.* Detailed visualization of thickness edges with $t(v_i)$ below the learned threshold $\tau$ (left) and those above $\tau$, filtered out by the thickness processor (right).

# H. Invariance/Equivariance Proof of the Proposed Data-driven Coordinate System

**Note:** Depending on whether the 'Inverse transformation' in Sec. 4.2.4 is applied or not, we can choose between E(3)-equivariance and invariance.

First, we will prove that our transformation algorithm $\mathbf{T}$ is invariant to translations $g \in \mathbb{R}^3$ and to rotations and reflections for any orthogonal matrix $Q \in \mathbb{R}^{3 \times 3}$. Specifically, we aim to show that the function $\mathbf{T}$ satisfies the following property, demonstrating its invariance. We will then discuss how applying the inverse transformation leads to equivariance.

$$\mathbf{x}^{\text{inv}} = \mathbf{T}(Q\mathbf{x}^{\text{orig}} + g) \tag{21}$$

We seek to demonstrate that $\mathbf{x}^{\text{inv}}$ is the same for any $Q$ and $g$, proving that the transformation $\mathbf{T}$ yields a consistent result regardless of the specific values of $Q$ and $g$. The transformation algorithm $\mathbf{T}$ consists of four main steps:

## Step 1: Center of Mass Adjustment

The center of mass of the original coordinates is given by (Eq. 3):

$$\mathbf{x}_{\text{cm}} = \frac{1}{|V|} \sum_{v_i \in V} \mathbf{x}_i^{\text{orig}}$$

The adjusted coordinates of the original points are then (Eq. 4):

$$\tilde{\mathbf{x}}_i = \mathbf{x}_i^{\text{orig}} - \mathbf{x}_{\text{cm}}$$

After applying the translation vector $g$ and the orthogonal matrix $Q$, the center of mass of the transformed coordinates becomes:

$$\mathbf{x}_{\text{cm},\mathbf{T}} = \frac{1}{|V|} \sum_{v_i \in V} (Q\mathbf{x}_i^{\text{orig}} + g) \tag{22}$$

Thus, the adjusted coordinates in the transformed space are:

$$\tilde{\mathbf{x}}_{i,\mathbf{T}} = Q\mathbf{x}_i^{\text{orig}} + g - \mathbf{x}_{\text{cm},\mathbf{T}} = Q\mathbf{x}_i^{\text{orig}} + g - \left( \frac{1}{|V|} \sum_{v_i \in V} (Q\mathbf{x}_i^{\text{orig}} + g) \right) = Q\tilde{\mathbf{x}}_i \tag{23}$$

Note that the translation vector $g$ cancels out in the final expression. This shows that the transformation is invariant to the translation $g$. Therefore, **after Step 1**, the adjusted coordinates $\tilde{\mathbf{x}}_i$ are unaffected by the translation $g$, and subsequent steps only consider the effects of the orthogonal matrix $Q$.

## Step 2: Principal Axis Generation

To generate the principal axes, we compute the covariance matrix $C_\mathbf{X}$ of the coordinate matrix $\mathbf{X}$ using the formula:

$$C_\mathbf{X} = \frac{1}{|V| - 1} \mathbf{X}^T \mathbf{X} \tag{24}$$

After applying the orthogonal matrix $Q$ to the coordinate matrix, the covariance matrix of the transformed coordinates $C_{Q\mathbf{X}}$ becomes:

$$C_{Q\mathbf{X}} = \frac{1}{|V| - 1} (Q\mathbf{X})^T (Q\mathbf{X}) = \frac{1}{|V| - 1} \mathbf{X}^T \mathbf{X} = C_\mathbf{X} \tag{25}$$

Thus, the orthogonal matrix does not affect the principal axis directions within the data; it only alters the basis. Let $\mathbf{R}^{\text{orig}}$ be the rotation matrix corresponding to the three principal basis vectors of the original coordinate matrix $\mathbf{X}^{\text{orig}}$, such that:

$$\mathbf{R}^{\text{orig}} = [\mathbf{b}_1, \mathbf{b}_2, \mathbf{b}_3] \in \mathbb{R}^{3 \times 3} \tag{26}$$

Then, the rotation matrix $\mathbf{R_T}$ for the adjusted coordinates becomes:

$$\mathbf{R_T} = Q\mathbf{R}^{\text{orig}} \tag{27}$$

**Step 3: Determining the Direction of Principal Axes**

It is important to note that the basis vectors generated by the PCA algorithm do not have an explicit sign convention, meaning a basis vector could be represented as $[1, -1, 1]$ or $[-1, 1, -1]$. To resolve this ambiguity, we introduce a data-driven criterion to determine the consistent sign of the basis vectors, such as using a reference vector that connects the center of the bounding box to the center of mass, as described in Section 4.1, Step 3.

While the proposed criterion in Step 3 generally yields consistent signs for the basis vectors across most real-world meshes, it has a limitation in cases where the shape exhibits perfect symmetry with respect to the three principal axes. Specifically, if the dot product between a principal axis $\mathbf{b}_i$ and the reference vector $\mathbf{v}$ is zero (i.e., $\mathbf{b}_i \cdot \mathbf{v} = 0$) in Eq. 5, the sign of that axis cannot be deterministically aligned. However, such perfectly symmetric meshes are rare in practice, especially for constructed real-world geometries.

For the proof, we assume that the signs of the basis vectors are already aligned.

**Step 4: Coordinate Transformation**

For the final transformation, we apply the rotation matrix $\mathbf{R}^{\text{orig}}$ to the adjusted coordinates $\tilde{\mathbf{x}}_i$ to obtain the data-driven invariant coordinate $\mathbf{x}^{\text{inv}}$:

$$\mathbf{x}^{\text{inv}} = \mathbf{R_T}^T \tilde{\mathbf{x}}_{i,\mathbf{T}} = (Q\mathbf{R}^{\text{orig}})^T Q\tilde{\mathbf{x}}_i = \mathbf{R}^{\text{orig}T} \tilde{\mathbf{x}}_i \tag{28}$$

Thus, $\mathbf{x}^{\text{inv}}$ is invariant to the translation vector $g$ and the orthogonal matrix $Q$. This shows that the transformation results in a data-driven invariant coordinate system that possesses E(3)-invariance.

**Final Prediction and Equivariance**

When we further process the 'inverse transformation' after decoding in Section 4.2.4, the final prediction $\mathbf{p}_i^{\text{inv}} \in \mathbb{R}^3$ in the invariant coordinate system is inverse-transformed as follows:

$$\mathbf{p}_i = \mathbf{R}\mathbf{p}_i^{\text{inv}} + \mathbf{x}_{\text{cm},\mathbf{T}} = (Q\mathbf{R}^{\text{orig}})(\mathbf{R}^{\text{orig}T}\tilde{\mathbf{p}}_i) + \mathbf{x}_{\text{cm},\mathbf{T}} = Q(\mathbf{p}^{\text{orig}} - \mathbf{x}_{\text{cm}}) + \frac{1}{|V|}\sum_{v_i \in V}(Q\mathbf{x}_i^{\text{orig}} + g) \tag{29}$$

This simplifies to:

$$\mathbf{p}_i = Q(\mathbf{p}^{\text{orig}} - \frac{1}{|V|}\sum_{v_i \in V}\mathbf{x}_i^{\text{orig}}) + \frac{1}{|V|}\sum_{v_i \in V}(Q\mathbf{x}_i^{\text{orig}} + g) \tag{30}$$

Thus, we have:

$$\mathbf{p}_i = Q\mathbf{p}^{\text{orig}} + g \tag{31}$$

This shows that the **final prediction preserves the E(3)-equivariance.** Therefore, depending on whether the 'inverse transformation' is applied or not, **we can choose between E(3)-equivariance and invariance.**

