# OpenReview forum: "Thickness-aware E(3)-Equivariant 3D Mesh Neural Networks"
_ICML.cc/2025/Conference — ICML 2025 poster_

### Official Review · Reviewer_pfVr · 2025-03-10

**Overall Recommendation:** 3

**Summary:**

This paper considers the thickness of the mesh when predicting the deformation. T-EMNN is proposed to preserve E(3)-equivariance and invariance. Experiments show that the proposed method outperforms baselines and the introduction of thickness benefits the baselines as well.

## Update after rebuttal
I appreciate the authors' efforts to address my concerns and I have updated my score to 3.

**Claims And Evidence:**

The claims are generally clear.

**Essential References Not Discussed:**

Since “E(3)-equivariance and invariance” is one of the core idea in this paper, the author may discuss neural network with the property of “E(3)-equivariance and invariance” in a separate paragraph. For example, the author may adjust L85-106 as part of the related work.

**Experimental Designs Or Analyses:**

The evaluations seem correct.

**Methods And Evaluation Criteria:**

My main concern is about the limited dataset. Is there any other dataset? More samples or different kinds of objects would strengthen this paper.

**Other Comments Or Suggestions:**

Please refer to the "Questions For Authors".

**Other Strengths And Weaknesses:**

The E(3)-equivariance is achieved by adjusting the inputs, while the GNN part does not contribute to this property. The proposed method mainly adopts the basic GNN module and introduces several conditional embeddings through concatenation. The main contributions mentioned above seem limited.

**Questions For Authors:**

I summarize my main concerns as follows:
1. Limited dataset. (Methods And Evaluation Criteria)
2. Please provide more details to emphasize the novelty of this paper. (Other Strengths And Weaknesses)
3. Reorganization of related work. (Essential References Not Discussed)

**Relation To Broader Scientific Literature:**

Please refer to "Essential References Not Discussed".

**Theoretical Claims:**

Please refer to "Claims And Evidence".

---

> ### Author Rebuttal · Authors · 2025-04-01
>
> Thank you for your thoughtful review. For a comprehensive response, please refer to the attached [link](https://shorturl.at/gOsz6). All materials within the link are indexed starting with the letter ‘L’ (e.g., Fig. L1).
>
> ---
>
> **A1. Novelty of Our Work.**
>
> We would like to organize our contributions into two key aspects: **1)** Thickness-Aware Framework and **2)** Data-driven Coordinates, which effectively achieve E(3)-equivariance. Given your concern about the novelty of the GNN architecture, ___we will focus on emphasizing the Thickness-Aware Framework in this rebuttal___, rather than the data-driven coordinates.
>
> Perhaps our architecture may appear simple, but we would like to emphasize that the ___input meshes do not inherently contain any information regarding the "thickness."___ As a result, a basic GNN without any thickness processing faces challenges in interacting between opposing surfaces, despite their high correlation, as shown in Fig. 1 in the introduction.
>
> Our novelty lies in how the GNN model ___adaptively addresses the "thickness,"___ even though this information is not provided in the input data. To achieve this, __we define the "thickness edge" from scratch and address potential issues__, such as the negative correlation between opposing surfaces when the thickness is too large to share the correlative reaction. To overcome this challenge, we propose a __learnable thickness threshold__ that adaptively identifies the optimal thickness edges. As a result, our method demonstrates strong performance, particularly on the $R^2$ metric, showcasing its ability to predict more realistic behavior.
>
> We would like to emphasize that our method is not distinguished from others simply by utilizing several conditional embeddings through concatenation. In fact, __the architecture of the baseline methods was modified to allow for the use of exactly the same conditional embeddings, including both condition and coordinate embeddings, as in our method, ensuring a fair comparison.__ Our distinct contribution lies in whether the model can handle opposing surfaces with high correlation, and our model successfully achieves this by defining and processing the thickness in a learnable way.
>
> ---
>
> **A2. Evaluation on Four Additional Datasets.**
>
> We evaluate our method on four additional datasets (using 5 seeds for reliability):
> - **Basket** (Graph-level prediction - Max pressure)
> - **Circular Plate** (Node-level prediction - Deflection of each node)
> - **SimJEB** [1] (Node-level prediction - Magnitude of the displacement)
>
> The dataset and results are presented in Table L2 and Figures L1-L5. In Table L2, our method demonstrates strong performance on both the Basket (graph-level prediction) and Circular Plate (node-level prediction) datasets. Regarding the SimJEB dataset, although our data-driven coordinate system does not perform well under the "In Dist." setting (where the original coordinate system is well-established but our system faces challenges in aligning coordinates between diverse shapes), it performs effectively in the "Out-of-Dist." setting. In this setting, our method successfully addresses the misalignment issue in the original coordinate system between shapes, ensuring E(3)-invariance.
>
> To qualitatively demonstrate our model’s ability to process thickness, we include the learned valid thickness edges in Fig. L4 and L5. As shown in these figures, our thickness processor effectively connects opposing surfaces (left), facilitating beneficial interactions, while filtering out opposing surfaces that negatively impact performance (right).
>
> - **Deforming Plate** (Node-level prediction - Position of Next timestep)
>
> Moreover, to evaluate the dynamic capabilities of our framework—specifically the thickness processor—we perform next-timestep deformation prediction using the **Deforming Plate** dataset (as described in the MGN paper) to demonstrate how our thickness processor helps the model handle dynamic scenarios. Since our data-driven coordinate system is designed for static analysis to prevent misalignment from affecting the results, we use the original coordinate system for dynamic analysis. The same mesh edge and node features as in the MGN paper, along with an additional node feature—the shortest world distance from the actuator to the node—are used to model interactions with the external environment. For detailed results, please refer to Table L1 and Fig. L6, where our model successfully accounts for the deforming plate’s thickness, leading to improved performance.
>
> ---
>
> **A3. Reorganization of related work.** Thank you for pointing this out. We will ensure that the related work regarding methods related to E(n)-equivariance/invariance is clearly addressed and reorganized separately.
>
> > [1] Whalen et al. "SimJEB: simulated jet engine bracket dataset." Computer Graphics Forum. Vol. 40. No. 5. 2021.

---

> > ### Comment · Reviewer_pfVr · 2025-04-03
> >
> > Thank you for your response and all my concerns have been addressed. I will raise my score to 3.

---

> > > ### Author Response · Authors · 2025-04-03
> > >
> > > Dear Reviewer pfVr,
> > >
> > > We are very happy to hear that your concerns have been resolved.
> > > Thank you for your constructive feedback, and we will ensure to include our rebuttal in the revised version if it is accepted.
> > >
> > > Best regards,
> > > The Authors

---

### Official Review · Reviewer_8wBp · 2025-03-10

**Overall Recommendation:** 4

**Summary:**

This paper presents the Thickness-aware E(3)-Equivariant Mesh Neural Network (T-EMNN), a novel graph neural network designed to efficiently integrate the thickness of 3D objects into mesh-based static analysis. The authors introduce an innovative thickness-aware framework that explicitly considers interactions between opposing surfaces through dedicated "thickness edges". The method employs a learned thickness threshold to distinguish genuine thickness relationships from irrelevant spatial connections (e.g., width) and uses data-driven coordinates to ensure E(3)-equivariance. The authors validate their method using a real-world industrial dataset focused on injection molding applications. T-EMNN significantly outperforms state-of-the-art graph-based methods in terms of prediction accuracy (RMSE, MAE, R²) while maintaining computational efficiency.

**Claims And Evidence:**

Good to me.

**Essential References Not Discussed:**

The reference is good to me

**Experimental Designs Or Analyses:**

The experimental designs, particularly the inclusion of in-distribution and out-of-distribution conditions, are sound. The ablation study (Fig. 6, Table 2) effectively demonstrates the contribution and optimization of the thickness threshold and thickness edge features, validating the robustness of the proposed methods.

**Methods And Evaluation Criteria:**

The evaluation criteria (RMSE, MAE, R²) and comparison methods (MGN, EGNN, EMNN) used by the authors are appropriate and clearly demonstrate the effectiveness of the thickness-aware design. Visualizations in Figure 7 illustrate clearly the error distributions and improvements achieved by T-EMNN. The visualization in Figure 9 show the effectiveness of the threshold. All of these are good. However, the practical utility of the method could be strengthened by demonstrating more clearly the application pipeline in industrial scenarios like injection molding.

**Other Comments Or Suggestions:**

Good to me.

**Other Strengths And Weaknesses:**

The proposed method effectively addresses a significant limitation of existing surface-based mesh methods by integrating thickness-awareness into mesh neural networks. Additionally, the proposed data-driven coordinate transformation notably enhances robustness against spatial transformations, as demonstrated by the clear performance improvements observed when integrating this component into existing baseline methods. The paper further provides comprehensive and thorough evaluations, including meaningful ablation studies that help clarify the contribution of each component.

However, the paper currently lacks demonstrations of practical applications beyond synthetic validation scenarios. Including concrete examples demonstrating direct applicability in real-world industrial tasks would significantly strengthen the paper. Furthermore, the application pipeline description could be more detailed to better bridge theoretical contributions and practical utility. Lastly, the paper would benefit from a discussion regarding potential extensions to handle non-watertight surfaces (i.e., surfaces with holes), as the current methodology does not appear to strictly require watertight constraints.

**Questions For Authors:**

1. Could you further elaborate on the practical integration of your model into an existing industrial workflow, particularly for injection molding? Would this integration introduce significant overhead?

2. How sensitive is the learned thickness threshold τ across significantly different geometries or scales, and how generalizable is this threshold to other applications beyond injection molding?

3. How to handle non-watertight inputs? For example, a "solid" surface with "small" holes no that. Does the proposed method robust to that kind of inputs?

**Relation To Broader Scientific Literature:**

The proposed work effectively builds upon and extends existing mesh-based neural network methods (MGN, EGNN, EMNN), clearly positioning itself within the broader scientific literature by addressing critical limitations related to thickness modeling and E(3)-Equivariance. It offers notable improvements in practical application contexts such as structural engineering and manufacturing.

**Theoretical Claims:**

Good to me

---

> ### Author Rebuttal · Authors · 2025-03-27
>
> We are grateful for your thorough review. For a comprehensive response, please refer to the attached [link](https://shorturl.at/gOsz6). All materials within the link are indexed starting with the letter ‘L’ (e.g., Fig. L1).
>
> ---
>
> **A1. Practical Utility of the Method (Response to Weakness and Q1).**
>
> The pipeline is illustrated in Fig. L10 to aid comprehension of the proposed approach. In the actual industrial process of injection molding product development, the process follows the stages of **design -> analysis -> mass production**. According to this process, designers must verify the product's defect status through analysis before mass production. However, **there is a bottleneck in the design -> analysis phase**, and we aim to address this using T-EMNN.
>
> The reason the design -> analysis process is a bottleneck is that the work is divided between the designer, who specializes in design, and the analyst, who specializes in numerical analysis. **The designer does not have numerical analysis expertise and must rely on the expert analyst.** This process requires iteration. When the designer modifies the design, they send it to the analyst for analysis. The analyst then reports the results to the designer, who evaluates whether there are any defects based on the analysis. If defects are suspected, the designer modifies the design and sends it again, creating a repeated cycle (design -> analysis -> design -> analysis). **This iterative process typically takes about a week for a single product** (based on the Basket dataset), leading to significant delays in mass production.
>
> We expect that using the **T-EMNN we developed will dramatically reduce this iterative process.** First, **T-EMNN allows designers to self-validate their designs** without requiring any special analysis skills. Furthermore, we confirmed that with **T-EMNN, it takes less than 3 minutes** (based on the Basket dataset) to go from preprocessing to checking the analysis results for a single injection molding product. Thus, with T-EMNN, designers can quickly self-validate designs that are unlikely to have defects. This allows designers to filter out designs with low defect potential and send only those to the analyst, significantly reducing the iteration process. This approach will drastically shorten the time to mass production.
>
> ---
>
> **A2. Generalizability of the Thickness Threshold (Response to Q2)**
>
> We evaluate our method on four additional settings (using 5 seeds for reliability):
> - **Deforming Plate** (Node-level prediction - Position of Next timestep)
> - **Basket** (Graph-level prediction - Max pressure)
> - **Circular Plate** (Node-level prediction - Deflection of each node)
> - **SimJEB** [1] (Node-level prediction - Magnitude of the displacement)
>
> Regarding performance, as shown in Table L1, the thickness processor successfully handles even next-timestep prediction. In Table L2, our method demonstrates strong performance on both the Basket (graph-level prediction) and Circular Plate (node-level prediction) datasets. Regarding the SimJEB dataset, although our data-driven coordinate system does not perform well under the "In Dist." setting (where the original coordinate system is well-established but our system faces challenges in aligning coordinates between diverse shapes), it performs effectively in the "Out-of-Dist." setting. In this setting, our method successfully addresses the misalignment issue in the original coordinate system between shapes, ensuring E(3)-invariance.
>
> To qualitatively demonstrate our model’s ability to process thickness, we include the learned valid thickness edges in Figs. L4, L5, and L6. As shown in these figures, our thickness processor effectively connects opposing surfaces (left), facilitating beneficial interactions, while filtering out opposing surfaces that negatively impact performance (right).
>
> ---
>
> **A3. Discussion for Handling Non-watertight Surfaces (Response to Q3)**
>
> We believe that **T-EMNN can be applied even in the case of non-watertight surfaces.** However, when creating the thickness edge, we utilize a method where a ray is cast from the starting node in the opposite direction of the normal (i.e., into the interior of the object) until it reaches the opposing surface. If the object is not watertight, there may be cases where the ray does not touch the opposite surface. In such cases, if the ray does not touch a specific node, the thickness edge can simply be excluded. This should not pose any issues for training and prediction.
>
> However, for our T-EMNN, the watertightness of the 3D shape data is not a critical factor. **What matters is whether the physical properties between thickness pairs are similar.** If they are similar, we expect that using T-EMNN will still yield good prediction results.

---

### Official Review · Reviewer_xjH9 · 2025-03-11

**Overall Recommendation:** 2

**Summary:**

This paper presents a novel graph neural network that incorporates thickness edges—connections between opposite sides of a surface mesh—to enable thickness-aware processing. To maintain E(3)-equivariance, it introduces a data-driven coordinate transformation. The model is evaluated on an injection molding dataset, where it outperforms various baselines.

**Claims And Evidence:**

There are two key issues with the claims and supporting evidence in the paper:

1. **Limited dataset evaluation** – The model is evaluated on a single novel dataset. While the results appear promising, testing on an additional dataset (or ideally two) would provide stronger support for the claimed superior performance. Given that the authors used MGN as a baseline, I wonder if they considered the *deforming plate* dataset, where a plate undergoes deformation over time. Applying their method to this dataset, particularly with a surface mesh representation, could further demonstrate its advantages.
2. **Unsubstantiated Equivariance Claim** – The paper states that the model “preserves E(3)-equivariance and invariance,” yet provides no theoretical proof to substantiate this claim. In my view, the lack of formal justification weakens the argument. Additionally, the phrasing raises a conceptual issue: how can the model be both equivariant and invariant? From my understanding, invariance is not merely a special case of equivariance, as it completely removes the output transformation rather than modifying it in a structured way. Instead, I would consider it a related but distinct concept. This imprecision in terminology further underscores the need for a more rigorous theoretical analysis to clarify and support the claim.

**Essential References Not Discussed:**

In Section 4.2.2 the authors propose their surface update. These are the exact update rules given in the Message Passing Neural network used for example by MeshGraphNet and originates from the paper "Graph Networks as Learnable Physics Engines for Inference and Control" by  Sanchez-Gonzalez et. al., 2018. I suggest to cite one of the papers and clearly distinguish here between own contribution and existing architectures.

**Experimental Designs Or Analyses:**

I encountered difficulties in fully assessing the experimental design due to ambiguities in the dataset's objectives and the lack of precise definitions for the evaluation metrics used. The metrics—Root Mean Squared Error (RMSE), Mean Absolute Error (MAE), and R-squared (R²)—are mentioned as abbreviations without explicit descriptions of their computation methods. While they are in general well-known, a precise defintion of them would strengthen the reproducibility of the paper's results.

Additionally, using 3 seeds for repetition is on the lower side, especially if only one dataset is used. Using confidence intervals instead of the standard deviation can also strenghten the statistical significance of the results.

**Methods And Evaluation Criteria:**

# Method
**Thickness Node Pair Definition and Mapping Consistency**

In Section 3.3, the paper introduces the concept of thickness node pairs, defined by a transformation $T$ that maps each node $v_i$ to another node on the opposite surface. However, the definition provided in Equation (1) does not ensure that applying the transformation twice returns the original node; that is, $T(T(v_i)) \neq v_i$. This issue becomes evident in materials where one surface is skewed relative to the other, resulting in a gradual thinning. The paper does not address how this scenario is handled, which raises concerns about the robustness of the thickness node pairing method in such cases. See for reference at the first draft I created for illustration: https://imgur.com/a/nmmI9oI

**Ray Projection Distance ($d$) as a Hyperparameter**

From my understanding, the ray projection distance $d$ is a crucial hyperparameter in defining thickness node pairs. In materials with multiple layers, determining the appropriate value of $d$ is essential to accurately pair nodes between surfaces. If the thickness varies between different sheets, selecting a single $d$ becomes challenging, potentially leading to incorrect thickness pairings. The paper does not discuss strategies for choosing ddd in such complex scenarios, leaving a gap in the methodology for materials with non-uniform thickness. In the second image provided in https://imgur.com/a/nmmI9oI , I don't see how the correct thickness pairs can be computed using a global value $d$.

**Selection of $\tau$**

I like the idea of selecting a data-driven $\tau$, but I am unsure if a global $\tau$ can suffice for non-uniform thickness objects. Going back to the second image in  https://imgur.com/a/nmmI9oI , a $\tau$ of 2 would exclude all thickness edges on the left side, while a $\tau$ of 6 would also include the "width" edges on the right side. Is my intuition here correct?

**Clarification of Node Features $g_i$ and $r_i$**

Table 1 references node features $g_i$ and $r_i$, but there is no explanation of these features in the main text or Appendix A. Clarifying the model inputs is important for understanding the model's functionality. Including descriptions of these features in the main body or referencing a specific section in the appendix would enhance the paper's clarity and comprehensibility.


**Thickness Activation Function**

Regarding the activation function for thickness pairs, the authors write "“while edges with $t(v_i) > \tau$ are excluded from the propagation process.” This is not precise, the activation function for $t(v_i) = \tau$ is 0.5, see the graph here: https://imgur.com/a/6Pyfzfr
As far as I understand it, the activation function has to be shifted to the left.

**Overall Assessment**

Despite these concerns, the proposed architecture for handling thickness appears sensible. Addressing the issues related to the transformation mapping, the selection of the ray projection distance $d$, and the clarification of node features would strengthen the paper's contributions and provide a more robust framework for thickness-aware processing in graph neural networks.

# Evaluation:

I did not fully understand the exact problem being addressed in the dataset. Unless I missed it, this information does not appear to be provided in the main text or in Appendix A. Figure 7a) presents a “Ground Truth” with an output value/vector(?) per node, but it is unclear what the referenced “deformation magnitude” represents.
What is the exact challenge of the dataset? What is given and what needs to be predicted?

Existing baselines, such as MGN, typically handle deformation tasks over time, addressing challenges like rollout stability by introducing noise. However, I suspect that this is not the case in the current task. If so, does the comparison to MGN only refer to its architecture and encoder-processor-decoder structure rather than its dynamic modeling capabilities?

In summary, the goal of the task is not clearly defined, and incorporating a dynamic deformation over time would provide a more appropriate basis for comparison with the baselines.

**Other Comments Or Suggestions:**

- 094, right: Mention MGN=”Mesh Graph Net”
- Inconsistent notation in Eq. 14: Normal $n_i$ uses only the node index $i$, while $n_{T(v_i)}$ uses the complete node $T(v_i)$ as an index.

**Other Strengths And Weaknesses:**

The general idea of improving the surface mesh by incorporating thickness edges is crucial for learnable physic simulators. I really like the idea and I can clearly see the scientific gap. The data-driven coordiante frame looks promising, I am wondering how this can be extended to multiple objects and their interaction, as well as how this is handling deformations over time. Also, what if gravity is a concern, does this not break the symmetry? Although this is a general problem with equivariant networks, it would strengthen the paper, if this is discussed.

**Questions For Authors:**

Main questions are discussed in "Methods And Evaluation Criteria".

**Relation To Broader Scientific Literature:**

The paper introduces a novel "Coordinate Transformation: E(3)-Invariant Data-driven Coordinate System" in section 4.1, which is not addressed in the related work section. This contribution could benefit from contextualization within the broader literature, especially related to coordinate transformations in geometric deep learning and invariant representations. Including references to works that explore coordinate transformations preserving E(3) invariance could strengthen the discussion.

Additionally, the paper mentions thickness handling, but only indirectly through mesh density and hierarchical pooling. It would be valuable to relate these ideas to prior work that explicitly addresses thickness handling in similar contexts. There may be relevant research on mesh processing or pooling methods that directly deals with the manipulation of thickness or geometric features in 3D data. If present, incorporating such references could provide a clearer connection to existing research and help position the paper's contributions more effectively in the broader field.

**Theoretical Claims:**

The paper does not provide theoretical proofs to support its claims. As mentioned earlier, the claim that the model “preserves E(3)-equivariance and invariance” lacks formal justification, making it difficult to assess its validity. A rigorous proof or at least a more detailed explanation would be necessary to substantiate this claim.

---

> ### Author Rebuttal · Authors · 2025-04-01
>
> Thank you for your thoughtful review. For a comprehensive response, please refer to the attached [link](https://shorturl.at/gOsz6). All materials within the link are indexed starting with the letter ‘L’ (e.g., Fig. L1).
>
> ---
>
> **A1. Problem Definition and Dynamic Analysis**
>
> The task is to predict node-wise 3D deformation for 28 unique shapes with thickness, each under 18 condition combinations. In Figure 7, we visualize the deformation by taking the L2-norm of each node's deformation, resulting in a scalar value per node. These shapes are tested under diverse conditions, which are defined as "static analysis," distinct from the dynamic analysis in MGN, which models material interactions. Static analysis focuses on how deformation changes under varying conditions (e.g., pressure), while dynamic analysis, such as the deforming plate in MGN, examines interactions over time (e.g., velocity).
>
> Although they serve different purposes, we also perform next timestep deformation prediction with the Deforming Plate dataset to demonstrate how our thickness processor helps the model handle dynamic situations. Since our data-driven coordinate system is designed for static analysis to prevent misalignment from affecting the results, we use the original coordinate system for dynamic analysis. The same mesh edge and node features as in the MGN paper, along with an additional node feature—the shortest world distance from the actuator to the node—are used to model interactions with the external environment. For detailed results, please refer to Table L1 and Fig. L6, where our model successfully handles the deforming plate's thickness, improving performance.
>
> ---
>
> **A2. Additional Datasets**
>
> Please refer to our _rebuttal A2 to reviewer "pfVr"_ due to character limits.
>
> ---
>
> **A3. Equivariance/Invariance Proof**
>
> Please refer to Sec. L1 for the detailed proof. Our model can be either invariant or equivariant, depending on whether the 'inverse transformation' is applied. We will make sure to be more rigorous in distinguishing between the terms "equivariance" and "invariance" in the revised version.
>
> ---
>
> **A4. Thickness Node Pair Definition.** Please refer to Fig. L7 and L8 in Sec. L3 for a comprehensive description. The dot product between the normal vectors of $v_i$ and $T(v_i)$ is included as an attribute of the thickness edge between $(v_i, T(v_i))$ to address the issue you raised. To better account for these non-perpendicular edges, we use the dot product to measure how well the normal vectors are aligned in the edge attribute.
>
> **A5. Ray Projection Distance ($d$).** We use the ray.intersects_location function from the trimesh library to find intersections between the mesh and the ray. Since the ray has no distance limit, it may extend to the far opposing side, leading to what is more accurately described as width rather than thickness, as shown in Fig. 4 of the paper. To address this, we introduce the thickness edge feature as a distance measure and apply a learnable thickness threshold to filter out incorrect thickness edges, without requiring a distance limit $d$.
>
> **A6. Selection of $\tau$.** The learnable threshold $\tau$ is essentially a thickness threshold that helps with the prediction. It defines the range of thickness edges where message passing is beneficial. As distance increases, the interaction or correlation between opposite nodes tends to decrease. This aligns with the design we intended, and the reason it is made learnable is to allow the model to dynamically adjust this threshold.
>
> **A7. Clarification of Node Features $g_i$  and $r_i$**  Node features are described in Apx, lines 606-607. We will reference them in the main text for clarity.
>
> **A8. Thickness Activation Function.** Please refer to Fig. L9 for details. The activation function in Eq. 15 includes $\alpha$, which controls transition sharpness. Increasing $\alpha$ sharpens the influence based on thickness, but the influence is gradually reduced, not abruptly cut off, to ensure stable training. For a detailed analysis of $\alpha$, see Apx. D. The term "exclude" was meant to indicate a gradual reduction, not a hard cutoff.
>
> **A9. Regarding MGN.** While our task does not require dynamic modeling, we chose MGN as a baseline for its strong architecture, on which we build by adding a thickness processor and a data-driven coordinate system. As stated in line 142, our method shares the core MGN structure, making a fair comparison and emphasizing the impact of our additions.
>
> ---
>
> **A10. Essential References Not Discussed**
>
> We would like to clarify that MGN is mentioned in line 141, where we note that our architecture is derived from the core MGN structure. However, for clarity, we will specifically reference it in Sec 4.2.2.
>
> ---
>
> **A11. Regarding the gravity.** Thank you for your insightful comment. We are indeed considering a coordinate system where equivariance operates beyond the direction of the gravitational field as part of our future work.

---

### Official Review · Reviewer_otWZ · 2025-03-14

**Overall Recommendation:** 3

**Summary:**

The paper introduces **Thickness-aware E(3)-Equivariant Mesh Neural Networks (T-EMNN)**, a framework designed to address the limitations of existing mesh-based 3D analysis methods, which often overlook the inherent thickness of real-world 3D objects. The authors argue that thickness plays a critical role in physical behaviors such as deformation, bending, and stress distribution, particularly in objects like plates, baskets, and layered materials. T-EMNN integrates thickness information into the mesh representation while maintaining computational efficiency and preserving E(3)-equivariance (invariance to translations, rotations, and reflections).
The paper also introduces a **thickness threshold** to dynamically regulate interactions between opposing surfaces, ensuring that only relevant thickness-related interactions are considered. Experimental results show that T-EMNN outperforms existing methods, including MGN, EGNN, and EMNN, in both in-distribution and out-of-distribution settings.

**Claims And Evidence:**

The claims made in the paper are well-supported by both theoretical and empirical evidence. However, the paper could benefit from a more detailed discussion of the limitations of the proposed method, particularly in scenarios where thickness is not the dominant factor in deformation.

**Essential References Not Discussed:**

It is ok for me.

**Experimental Designs Or Analyses:**

The experimental design is robust, with the following key strengths: Dataset, Baselines, Ablation studies, Out-of-distribution testing. One potential limitation is the lack of comparison with volume-based methods (DetailRecon: Focusing on Detailed Regions for Online Monocular 3D Reconstruction), which could provide additional context for the performance of surface-based approaches like T-EMNN.

**Methods And Evaluation Criteria:**

The proposed methods are well-suited for the problem at hand. The experimental design is sound, with a clear comparison to state-of-the-art baselines (MGN, EGNN, EMNN) and ablation studies to validate the contributions of individual components (e.g., thickness edges, dot product of normal vectors).
Could the proposed show more visual examples on the thickness reconstruction for challenging mesh, such as the leaves, cloth, etc.

**Other Comments Or Suggestions:**

NA

**Other Strengths And Weaknesses:**

**Strengths**:
1. The introduction of thickness-aware modeling and data-driven coordinates is a significant contribution to the field of 3D mesh analysis.
2. The validation on a real-world industrial dataset demonstrates the method's applicability to practical problems like injection molding.
3. The model's performance under out-of-distribution settings highlights its robustness to transformations.

**Weaknesses**:
1. The paper focuses on surface meshes but does not compare T-EMNN with volume-based methods, which are commonly used in structural analysis.
2. While the paper provides a clear explanation of the proposed methods, formal theoretical proofs could strengthen the theoretical contributions.

**Questions For Authors:**

1. How does T-EMNN compare to volume-based methods (like FEM, DetailRecon: Focusing on Detailed Regions for Online Monocular 3D Reconstruction) in terms of accuracy and computational efficiency?
2. Can the authors provide formal proofs for the E(3)-equivariance and invariance properties of the proposed data-driven coordinate system?
3. How well does T-EMNN generalize to other types of 3D objects beyond the industrial dataset used in the experiments?

**Relation To Broader Scientific Literature:**

The paper builds on several key areas of research, Mesh-based 3D representation, E(3)-equivariant networks, and Thickness modeling.
The paper situates itself well within the broader context of 3D mesh analysis and equivariant neural networks, but it could benefit from a more detailed discussion of how T-EMNN compares to volume-based methods, which are commonly used in structural analysis.

**Theoretical Claims:**

The paper does not present formal theoretical proofs but provides a clear explanation of the E(3)-equivariance and invariance properties of the proposed data-driven coordinate system. The authors justify their approach using geometric principles and demonstrate its effectiveness empirically.

---

> ### Author Rebuttal · Authors · 2025-04-01
>
> Thank you for taking the time to review our work. For a comprehensive response, please refer to the attached [link](https://shorturl.at/gOsz6). All materials within the link are indexed starting with the letter ‘L’ (e.g., Fig. L1).
>
> ---
>
> **A1. Additional Datasets for Challenging Meshes (Response to Methods And Evaluation Criteria and W3)**
>
> We evaluate our method on four additional datasets (using 5 seeds for reliability):
> - **Deforming Plate** (Node-level prediction - Position of Next timestep)
> - **Basket** (Graph-level prediction - Max pressure)
> - **Circular Plate** (Node-level prediction - Deflection of each node)
> - **SimJEB** [1] (Node-level prediction - Magnitude of the displacement)
>
> **To visualize the thickness edges on other challenging datasets**, we include the learned valid thickness edges in Figs. L4, L5, and L6. As shown in these figures, our thickness processor effectively connects opposing surfaces (left), facilitating beneficial interactions, while filtering out opposing surfaces that negatively impact performance (right).
>
> **Regarding performance**, as shown In Table L2, our method demonstrates strong performance on both the Basket (graph-level prediction) and Circular Plate (node-level prediction) datasets. Regarding the SimJEB dataset, although our data-driven coordinate system does not perform well under the "In Dist." setting (where the original coordinate system is well-established but our system faces challenges in aligning coordinates between diverse shapes), it performs effectively in the "Out-of-Dist." setting. In this setting, our method successfully addresses the misalignment issue in the original coordinate system between shapes, ensuring E(3)-invariance.
>
> Moreover, to evaluate the **dynamic capabilities of our framework**—specifically the thickness processor—we perform next-timestep deformation prediction using the Deforming Plate dataset (as described in the MGN paper) to demonstrate how our thickness processor helps the model handle dynamic scenarios. Since our data-driven coordinate system is designed for static analysis to prevent misalignment from affecting the results, we use the original coordinate system for dynamic analysis. The same mesh edge and node features as in the MGN paper, along with an additional node feature—the shortest world distance from the actuator to the node—are used to model interactions with the external environment. For detailed results, please refer to Table L1, where our model successfully accounts for the deforming plate’s thickness, leading to improved performance.
>
> ---
>
> **A2. Discussion on Limitations in Scenarios Where Thickness is Not Dominant for Prediction (Response to Claims And Evidence)**
>
> Since not all objects have thickness, our method may not be very helpful for such shapes. However, in these cases, we adjust the threshold where thickness influences performance in a learnable manner. In such situations, **the thickness threshold will adapt so that no thickness edge contributes to the prediction.** If thickness is not important, the thickness edge will be ignored, and the model will ultimately be equivalent to the MGN with our proposed data-driven coordinate architecture.
>
> ---
>
> **A3. Equivariance/Invariance Proof (Response to Theoretical Claims, W2 and Q2)**
>
> Please refer to Sec. L1 for the detailed proof. Our model can be either invariant or equivariant, depending on whether the 'inverse transformation' is applied.
>
> ---
>
> **A4. Surface Mesh vs. Volume Mesh (Response to W1 and Q1)**
>
> Please refer to Appendix G for a comparison of GPU memory usage, performance, and inference time between volume-based and surface-based methods. We use the same model architecture (MGN) for a fair comparison, with the distinction that the input data is either volume mesh or surface mesh. The surface mesh utilizes only 11% of the GPU memory and takes 19% of the inference time compared to the volume mesh, while also delivering better performance. We argue that the performance degradation of the volume mesh arises from the over-smoothing or over-squashing problem that occurs when the mesh density is too high in GNN-based methods. For more details, please refer to Appendix G.
>
> Regarding the DetailRecon, which was published in January 2025, we did not recognize the paper, but we will address it in the related work section of the revised version. Moreover, it is worth noting that FEM is not a neural network-based method, but rather a traditional numerical technique used for solving partial differential equations. While FEM remains a cornerstone in engineering simulations, it differs fundamentally from neural network-based approaches like ours, which aim to learn patterns from data rather than relying solely on predefined physical models.

---

> > ### Comment · Reviewer_otWZ · 2025-04-03
> >
> > Thanks for the detailed response to my concerns, especially for the additional results. The attached results have addressed my concerns and still encourage the authors to discuss and show some failure cases (it would be better). I still maintain our initial scores.

---

### Decision · Program_Chairs · 2025-05-01

**Decision:**

Accept (poster)

**Comment:**

The authors introduce a thickness-aware variant of E(3)-equivariant mesh neural networks, integrating surface thickness into mesh processing frameworks. The paper was well-received by the reviewers, resulting in three positive (2x WA, 1x A) and one WR recommendation. The reviewers welcomed the introduction of thickness as a contribution of high practical significance and highlighted its effectiveness and robustness. Concerns remained regarding the number of evaluated real datasets as well as questions regarding clarity in equi-and invariances. The authors provided answers to the individual concerns, convincing all but one reviewer.

I will follow with an accept recommendation. While one reviewer remained with a negatively leaning score, I believe his strongest concerns (regarding equi-and invariance) have been addressed sufficiently by the authors.